# Agnostically Learning Single-Index Models using Omnipredictors

**Aravind Gollakota**
Apple

**Parikshit Gopalan**
Apple

**Adam R. Klivans**
UT Austin

**Konstantinos Stavropoulos**
UT Austin

## Abstract

We give the first result for agnostically learning Single-Index Models (SIMs) with arbitrary monotone and Lipschitz activations. All prior work either held only in the realizable setting or required the activation to be known. Moreover, we only require the marginal to have bounded second moments, whereas all prior work required stronger distributional assumptions (such as anticoncentration or boundedness). Our algorithm is based on recent work by Gopalan et al. [2023] on omniprediction using predictors satisfying calibrated multiaccuracy. Our analysis is simple and relies on the relationship between Bregman divergences (or matching losses) and $\ell_p$ distances. We also provide new guarantees for standard algorithms like GLMtron and logistic regression in the agnostic setting.

## 1 Introduction

Generalized Linear Models (GLMs) and Single-Index Models (SIMs) constitute fundamental frameworks in statistics and supervised learning [McCullagh, 1984, Agresti, 2015], capturing and generalizing basic models such as linear and logistic regression. In the GLM framework, labeled examples $(\mathbf{x}, y)$ are assumed to satisfy $u(\mathbb{E}[y|\mathbf{x}]) = \mathbf{w} \cdot \mathbf{x}$ (or $\mathbb{E}[y|\mathbf{x}] = u^{-1}(\mathbf{w} \cdot \mathbf{x})$), where $u$ is a known monotone function (called the link function) and $\mathbf{w}$ is an unknown vector. Single-Index Models (SIMs) are a generalization in which the monotone link function $u$ is also unknown.

In the realizable setting where the labels are indeed generated according to a GLM with a known Lipschitz link function, the GLMtron algorithm of Kakade et al. [2011] is a simple and efficient learning algorithm. When the ground truth is only assumed to be a SIM (i.e. the link function is unknown), it can be learned efficiently by the Isotron algorithm [Kalai and Sastry, 2009, Kakade et al., 2011].

In this work, we consider the significantly more challenging *agnostic* setting, where the labels are arbitrary and not necessarily realizable by any SIM. Importantly, we do not fix a link function in advance; our goal is to output a predictor that has squared error comparable to that of the optimal SIM with an arbitrary monotone and Lipschitz link function. We can equivalently view this as a natural squared-error regression problem in which the final optimality guarantee must hold with respect to all SIMs with bounded weights and monotone, Lipschitz link functions.[1]

Formally, consider a distribution $D$ over $\mathbb{R}^d \times [0, 1]$ and denote the squared error of a function $h : \mathbb{R}^d \to \mathbb{R}$ by $\mathrm{err}_2(h) = \mathbb{E}_{(\mathbf{x},y) \sim D}[(y - h(\mathbf{x}))^2]$. Let $\mathrm{opt}(\mathrm{SIM})$ denote optimal value of $\mathrm{err}_2(h)$ over all SIMs $h$ with bounded weights and arbitrary 1-Lipschitz monotone activations (we call the inverse $u^{-1}$ of a link function $u$ the activation function). Given access to samples from $D$, the goal of an agnostic learning algorithm is to compute a predictor $p : \mathbb{R}^d \to [0, 1]$ with error $\mathrm{err}_2(p)$ that, with high probability over the samples, is comparable to the error of the optimal SIM:

$$\mathrm{err}_2(p) \leq \mathrm{opt}(\mathrm{SIM}) + \varepsilon.$$

---

[1] In this context, in recent times it has also been common to refer to the class of GLMs (resp. SIMs) as the class of *single neurons* with known (resp. unknown) activation functions.

37th Conference on Neural Information Processing Systems (NeurIPS 2023).

Our main result is the first efficient learning algorithm with a guarantee of this form.

**Theorem 1.1** (Informal, see Theorem 3.1). *Let* $\mathrm{SIM}_B$ *denote the class of SIMs of the form* $\mathbf{x} \mapsto u^{-1}(\mathbf{w} \cdot \mathbf{x})$ *for some* 1-*Lipschitz function* $u^{-1}$ *and* $\|\mathbf{w}\|_2 \leq B$. *Let* $D$ *be any distribution over* $\mathbb{R}^d \times [0, 1]$ *whose marginal on* $\mathbb{R}^d$ *has bounded second moments. There is an efficient algorithm (Algorithm 1) that agnostically learns* $\mathrm{SIM}_B$ *over* $D$ *up to error*

$$\mathrm{err}_2(p) \leq O\big(B\sqrt{\mathrm{opt}(\mathrm{SIM}_B)}\big) + \varepsilon.$$

This result provides a guarantee comparable to that of the Isotron algorithm [Kalai and Sastry, 2009, Kakade et al., 2011] (which also tackles the SIM setting, where the link function is unknown) but for the challenging agnostic setting rather than the realizable setting (where $\mathrm{opt}(\mathrm{SIM}_B, D) = 0$). Moreover, Isotron's guarantees require the distribution to be supported on the unit ball, whereas we only require a mild second moment condition.

In our view, this result helps establish Algorithm 1 (and indeed any algorithm to compute omnipredictors, as we introduce and discuss shortly) as an efficient and powerful *off-the-shelf* supervised learning algorithm, akin to random forests or gradient-boosted trees.

A natural question is whether our guarantees are near-optimal, e.g., whether we can obtain a guarantee of the form $\mathrm{err}_2(p) \leq \mathrm{opt}(\mathrm{SIM}) + \varepsilon$. However, there is strong evidence that such results cannot be obtained using efficient algorithms [Goel et al., 2019, Diakonikolas et al., 2020a, Goel et al., 2020, Diakonikolas et al., 2021]. We partially justify the form of our guarantee by showing in Section 5 (adapting a result due to Diakonikolas et al. [2022a]) that one cannot avoid a dependence on the norm bound $B$. Further closing the gap between upper and lower bounds is an important direction for future work.

**Overview of techniques.**    We now describe the main ingredients and techniques that go into proving Theorem 1.1. Our starting point is the connection between GLMs and so-called matching losses [Auer et al., 1995]. This connection arises from the fact that fitting a GLM with a known link function is equivalent to minimizing (over the class of all linear functions) a certain convex loss known as the matching loss corresponding to that link function (see Definition 1.8). (We refer the reader to [Gopalan et al., 2023, Sec 5.1] for a more detailed discussion.)

Importantly for us, minimizing the matching loss corresponding to a link function $u$ yields a meaningful guarantee even in the agnostic setting, where the Bayes optimal predictor (i.e. $\mathbf{x} \mapsto \mathbb{E}[y|\mathbf{x}]$) is not necessarily a GLM at all. Specifically, it turns out to be equivalent (via Fenchel-Legendre duality) to finding the closest predictor to the Bayes optimal predictor in the metric of the *Bregman divergence* associated with the link function $u$ (see e.g. [Gopalan et al., 2023, Lemma 5.4]).

This is a powerful observation, but it still assumes that we have a fixed and known link function $u$ in mind. In the agnostic setting, it is arguably unrealistic to assume that we know the best link function for the distribution at hand. Remarkably, recent work by Gopalan et al. [2022, 2023] has shown that there exist efficient learning algorithms that simultaneously minimize all matching losses corresponding to arbitrary monotone and Lipschitz link functions. Their solution concept is called an *omnipredictor*, i.e., a single predictor that is able to compete with the best-fitting classifier in a class $\mathcal{C}$ as measured by a large class of losses (as opposed to just a single pre-specified loss, as is standard in machine learning). They obtain such predictors through calibrated multiaccuracy [Gopalan et al., 2023] or multicalibration [Gopalan et al., 2022].

From the point of view of ordinary supervised learning or regression, however, an optimality guarantee in terms of such matching losses or Bregman divergences is hard to interpret. A much more standard metric is simply that of squared error. The key final step for our results is to find a way of translating (a) optimality (ranging over all linear functions) in terms of all matching losses simultaneously into (b) optimality (ranging over all SIMs) in terms of squared error. We do so by proving simple analytic *distortion inequalities* relating matching losses to $\ell_p$ losses, which we believe may be of independent interest.

On a technical level, to prove these distortion inequalities we first prove strong versions of such inequalities for matching losses arising from bi-Lipschitz link functions (see Lemma 2.2). We then obtain our results for general Lipschitz link functions by carefully approximating them using bi-Lipschitz link functions (see Lemma 3.3).

**Further applications.** As further applications of our approach, if we let $\mathrm{opt}(\mathrm{GLM}_{u^{-1},B})$ denote the optimal value of $\mathrm{err}_2(h)$ over all GLMs of the form $\mathbf{x} \mapsto u^{-1}(\mathbf{w} \cdot \mathbf{x})$, where $\|\mathbf{w}\|_2 \leq B$, we obtain the following result about bi-Lipschitz link functions (including, for example, the Leaky ReLU).

**Theorem 1.2** (Informal, see Theorem 2.1). *Let $u : \mathbb{R} \to \mathbb{R}$ be a bi-Lipschitz invertible link function. Then, any predictor $p : \mathbb{R}^d \to \mathbb{R}$ that is an $\varepsilon$-approximate minimizer of the population matching loss that corresponds to $u$, with respect to a distribution $D$ over $\mathbb{R}^d \times [0,1]$ satisfies*

$$\mathrm{err}_2(p) \leq O\big(\mathrm{opt}(\mathrm{GLM}_{u^{-1},B})\big) + O(\varepsilon)$$

This guarantee holds under milder distributional assumptions than are required by comparable prior work on agnostically learning GLMs or single neurons [Frei et al., 2020, Diakonikolas et al., 2022b]. Moreover, when we focus on distortion bounds between the logistic loss and the squared loss, we obtain a near-optimal guarantee of $\widetilde{O}(\mathrm{opt}(\mathrm{GLM}_{u^{-1},B}))$ for logistic regression, when $u$ is the logit link function (i.e., when $\mathrm{GLM}_{u^{-1},B}$ is the class of sigmoid neurons).

**Theorem 1.3** (Informal, see Theorem 4.1). *Let $u(t) = \ln(\frac{t}{1-t})$. Then, any predictor $p : \mathbb{R}^d \to \mathbb{R}$ that is an approximate $\varepsilon$-minimizer of the population logistic loss, with respect to a distribution $D$ over $\mathbb{R}^d \times [0,1]$ whose marginal on $\mathbb{R}^d$ has subgaussian tails in every direction satisfies*

$$\mathrm{err}_2(p) \leq \widetilde{O}\big(\mathrm{opt}(\mathrm{GLM}_{u^{-1},B})\big) + O(\varepsilon)$$

While our error guarantee for this problem is weaker than that of Diakonikolas et al. [2022b], we do not make the anti-concentration assumptions their results require.

## 1.1 Background and Relation to Prior Work

We note that matching losses have been studied in various previous works either implicitly [Kakade et al., 2011] or explicitly [Auer et al., 1995, Diakonikolas et al., 2020b, Gopalan et al., 2023] and capture various fundamental algorithms like logistic and linear regression [McCullagh, 1984, Agresti, 2015]. However, to the best of our knowledge, our generic and direct approach for transforming matching loss guarantees to squared error bounds, has not been explored previously. Furthermore, our results do not depend on the specific implementation of an algorithm, but only on the matching loss bounds achieved by its output. In this sense, we provide new agnostic error guarantees for various existing algorithms of the literature. For example, our results imply new guarantees for the GLMtron algorithm of Kakade et al. [2011] in the agnostic setting, since GLMtron can be viewed as performing gradient descent (with unit step size) on the matching loss corresponding to a specified link function.

Matching losses over linear functions are also linked to the Chow parameters [O'Donnell and Servedio, 2008] through their gradient with respect to $\mathbf{w}$, as observed by Diakonikolas et al. [2020b]. In fact, the norm of the matching loss gradient is also linked to multiaccuracy, a notion that originates to fairness literature [Hebert-Johnson et al., 2018, Kim et al., 2019]. A stationary point $\mathbf{w}$ of a matching loss that corresponds to a GLM with link $u$ is associated with a multiaccurate predictor $p(\mathbf{x}) = u^{-1}(\mathbf{w} \cdot \mathbf{x})$, i.e., a predictor such that for all $i \in [d]$, $\mathbb{E}[\mathbf{x}_i(y - p(\mathbf{x}))] = 0$. The work of [Gopalan et al., 2022, 2023] on omnipredictors presents a single predictor that is better than any linear model $\mathbf{w} \cdot \mathbf{x}$ for every matching loss. The results of Gopalan et al. [2022] show that a multicalibrated predictor (with respect to the features $\mathbf{x}_i$) is an omnipredictor for all convex losses, whereas Gopalan et al. [2023] shows that the simpler condition of calibrated multiaccuracy suffices for matching losses that arise from GLMs. In view of the relationship between multiaccuracy and the gradient of the matching loss, our results show that, while multiaccuracy implies bounds on agnostically learning GLMs, the additional guarantee of calibration is sufficient for agnostically learning all SIMs.

The work of Shalev-Shwartz et al. [2011] showed strong agnostic learning guarantees in terms of the absolute error (rather than the squared error) of the form $\mathrm{opt} + \varepsilon$ for a range of GLMs, but their work incurs an exponential dependence on the weight norm $B$. In comparison, for the absolute loss, we get a bound of the form $B\,\mathrm{opt}\log(1/\mathrm{opt})$ for logistic regression (see Theorem 4.3). In more recent years, the problem of agnostically learning GLMs has frequently also been phrased as the problem of agnostically learning single neurons (with a known activation). For the ReLU activation, work by Goel et al. [2017] showed an algorithm achieving error $\mathrm{opt} + \varepsilon$ in time $\mathrm{poly}(d)\exp(1/\varepsilon)$ over marginals on the unit sphere, and Diakonikolas et al. [2020b] showed an algorithm achieving error

$O(\mathrm{opt}) + \varepsilon$ in fully polynomial time over isotropic log-concave marginals. The work of Frei et al. [2020], Diakonikolas et al. [2022b] both show guarantees for learning general neurons (with known activations) using the natural approach of running SGD directly on the squared loss (or a regularized variant thereof). Frei et al. [2020] achieves error $O(\mathrm{opt})$ for any given strictly increasing activation and $O(\sqrt{\mathrm{opt}})$ for the ReLU activation, but they assume that the marginal distribution is bounded. In contrast, we only assume that the marginal distribution has bounded second moments and we do not consider that the activation is known. Diakonikolas et al. [2022b] proved an $O(\mathrm{opt})$ guarantee for a wide range of activations (including the ReLU), in the setting where the activation is known and over a large class of structured marginals, which need, however, to satisfy strong concentration and anti-concentration properties.

In terms of lower bounds and hardness results for this problem, the work of [Goel et al., 2019, Diakonikolas et al., 2020a, Goel et al., 2020, Diakonikolas et al., 2021, 2022a] has established superpolynomial hardness even for the setting of agnostically learning single ReLUs over Gaussian marginals.

**Limitations and directions for future work.** While we justify a certain dependence on the norm bound $B$ in our main result on agnostically learning SIMs, we do not completely justify the exact form of Theorem 1.1. An important direction for future work is to tightly characterize the optimal bounds achievable for this problem, as well as to show matching algorithms.

## 1.2 Preliminaries

For the following, $(\mathbf{x}, y)$ is used to denote a labelled sample from a distribution $D$ over $\mathbb{R}^d \times \mathcal{Y}$, where $\mathcal{Y}$ denotes the interval $[0, 1]$ unless it is specified to be the set $\{0, 1\}$. We note that, although we provide results for the setting where the labels lie within $[0, 1]$, we may get similar results for any bounded label space. We use $\mathbb{P}_D$ (resp. $\mathbb{E}_D$) to denote the probability (resp. expectation) over $D$ and, similarly, $\mathbb{P}_S$ (resp. $\mathbb{E}_S$) to denote the corresponding empirical quantity over a set $S$ of labelled examples. Throughout the paper, we will use the term differentiable function to mean a function that is differentiable except on finitely many points. Our main results will assume the following about the marginal distribution on $\mathbb{R}^d$.

**Definition 1.4** (Bounded moments). For $\lambda \geq 1$ and $k \in \mathbb{N}$, we say that a distribution $D_{\mathbf{x}}$ over $\mathbb{R}^d$ has $\lambda$-bounded $2k$-th moments if for any $\mathbf{v} \in \mathbb{S}^{d-1}$ we have $\mathbb{E}_{\mathbf{x} \sim D_{\mathbf{x}}}[(\mathbf{v} \cdot \mathbf{x})^{2k}] \leq \lambda$.

For a concept class $\mathcal{C} : \mathbb{R}^d \to \mathbb{R}$, we define $\mathrm{opt}(\mathcal{C}, D)$ to be the minimum squared error achievable by a concept $c : \mathbb{R}^d \to \mathbb{R}$ in $\mathcal{C}$ with respect to the distribution $D$. We now define our main learning task.

**Definition 1.5** (Agnostic learning). Let $\mathcal{C} : \mathbb{R}^d \to \mathbb{R}$ be a concept class, let $\Psi : [0, 1] \to [0, 1]$ be an increasing function, let $D$ be a distribution over $\mathbb{R}^d \times [0, 1]$ and $\varepsilon > 0$. We say that an algorithm $\mathcal{A}$ agnostically learns the class $\mathcal{C}$ up to error $\Psi(\mathrm{opt}(\mathcal{C}, D)) + \varepsilon$ if algorithm $\mathcal{A}$, given a number of i.i.d. samples from $D$, outputs, with probability at least $2/3$ over the samples and the randomness of $\mathcal{A}$, a hypothesis $h : \mathbb{R}^d \to [0, 1]$ with $\mathbf{E}_D[(y - h(\mathbf{x}))^2] \leq \Psi(\mathrm{opt}(\mathcal{C}, D)) + \varepsilon$.

We will also provide results that are specific to the sigmoid activation and work under the assumption that the marginal distribution is sufficiently concentrated.

**Definition 1.6** (Concentrated marginals). For $\lambda > 0$ and $\gamma$, we say that a distribution $D_{\mathbf{x}}$ over $\mathbb{R}^d$ is $(\lambda, \gamma)$-concentrated if for any $\mathbf{v} \in \mathbb{S}^{d-1}$ and $r \geq 0$ we have $\mathbb{P}_{\mathbf{x} \sim D_{\mathbf{x}}}[|\mathbf{v} \cdot \mathbf{x}| \geq r] \leq \lambda \cdot \exp(-r^\gamma)$.

**Definition 1.7** (Fenchel-Legendre pairs). We call a pair of functions $(f, g)$ a Fenchel-Legendre pair if the following conditions hold.

1. $g' : \mathbb{R} \to \mathbb{R}$ is continuous, non-decreasing, differentiable and 1-Lipschitz with range $\mathrm{ran}(g') \supseteq (0, 1)$ and $g(t) = \int_0^t g'(\tau) \, d\tau$, for any $t \in \mathbb{R}$.

2. $f : \mathrm{ran}(g') \to \mathbb{R}$ is the convex conjugate (Fenchel-Legendre transform) of $g$, i.e., we have $f(r) = \sup_{t \in \mathbb{R}} r \cdot t - g(t)$ for any $r \in \mathrm{ran}(g')$.

For such pairs of functions (and their derivatives $f', g'$), the following are true for $r \in \mathrm{ran}(g')$ and $t \in \mathrm{ran}(f')$ (note that $\mathrm{ran}(f')$ is not necessarily $\mathbb{R}$ when $g'$ is not invertible).

$$g'(f'(r)) = r \quad \text{and} \quad f(r) = rf'(r) - g(f'(r)), \text{ for } r \in \mathrm{ran}(g') \tag{1.1}$$

$$f'(g'(t)) = t \quad \text{and} \quad g(t) = tg'(t) - f(g'(t)), \text{ for } t \in \mathrm{ran}(f') \tag{1.2}$$

Note that $g'$ will be used as an activation function for single neurons and $f'$ corresponds to the unknown link function of a SIM (or the known link function of a GLM). We say that $g'$ is bi-Lipschitz if for any $t_1 < t_2 \in \mathbb{R}$ we have that $(g'(t_2) - g'(t_1))/(t_2 - t_1) \in [\alpha, \beta]$. If $g'$ is $[\alpha, \beta]$ bi-Lipschitz, then $f'$ is $[\frac{1}{\beta}, \frac{1}{\alpha}]$ bi-Lipschitz. However, the converse implication is not necessarily true when $g'$ is not strictly increasing.

**Definition 1.8** (Matching Losses). For a non-decreasing and Lipschitz activation $g' : \mathbb{R} \to \mathbb{R}$, the matching loss $\ell_g : \mathcal{Y} \times \mathbb{R} \to \mathbb{R}$ is defined pointwise as follows:

$$\ell_g(y, t) = \int_0^t g'(\tau) - y \, d\tau,$$

where $g(t) = \int_0^t g'$. The function $\ell_g$ is convex and smooth with respect to its second argument. The corresponding population matching loss is

$$\mathcal{L}_g(c \,; D) = \mathop{\mathbb{E}}_{(\mathbf{x}, y) \sim D} \Big[ \ell_g(y, c(\mathbf{x})) \Big] \tag{1.3}$$

In Equation (1.3), $c : \mathbb{R}^d \to \mathbb{R}$ is some concept and $D$ is some distribution over $\mathbb{R}^d \times [0, 1]$. In the specific case where $c$ is a linear function, i.e., $c(\mathbf{x}) = \mathbf{w} \cdot \mathbf{x}$, for some $\mathbf{w} \in \mathbb{R}^d$, then we may alternatively denote $\mathcal{L}_g(c \,; D)$ with $\mathcal{L}_g(\mathbf{w} \,; D)$.

We also define the Bregman divergence associated with $f$ to be $\mathbb{D}_f(q, r) = f(q) - f(r) - (q - r)f'(r)$, for any $q, r \in \mathrm{ran}(g')$. Note that $\mathbb{D}_f(q, r) \geq 0$ with equality iff $q = r$.

**Definition 1.9** (SIMs and GLMs as Concept Classes). For $B > 0$, we use $\mathrm{SIM}_B$ to refer to the class of all SIMs of the form $\mathbf{x} \mapsto g'(\mathbf{w} \cdot \mathbf{x})$ where $\|\mathbf{w}\|_2 \leq B$ and $g' : \mathbb{R} \to \mathbb{R}$ is an arbitrary 1-Lipschitz monotone activation that is differentiable (except possibly at finitely many points). We define $\mathrm{GLM}_{g', B}$ similarly except for the case where $g'$ is fixed and known.

We also define the notion of calibrated multiaccuracy that we need to obtain omnipredictors in our context.

**Definition 1.10** (Calibrated Multiaccuracy). A predictor $p : \mathbb{R}^d \to [0, 1]$ is called $\varepsilon$-multiaccurate if for all $i \in [d]$, $|\mathbb{E}[\mathbf{x}_i(y - p(\mathbf{x}))]| \leq \varepsilon$. It is called $\varepsilon$-calibrated if $|\mathbb{E}_{p(\mathbf{x})} \mathbb{E}_{y|p(\mathbf{x})}[y - p(\mathbf{x})]| \leq \varepsilon$.

## 2 Distortion Bounds for the Matching Loss

In this section, we propose a simple approach for bounding the squared error of a predictor that minimizes a (convex) matching loss, in the agnostic setting. We convert matching loss bounds to squared loss bounds in a generic way, through appropriate pointwise distortion bounds between the two losses. In particular, for a given matching loss $\mathcal{L}_g$, we transform guarantees on $\mathcal{L}_g$ that are competitive with the optimum linear minimizer of $\mathcal{L}_g$ to guarantees on the squared error that are competitive with the optimum GLM whose activation ($g'$) depends on the matching loss at hand.

We now provide the main result we establish in this section.

**Theorem 2.1** (Squared Error Minimization through Matching Loss Minimization). *Let $D$ be a distribution over $\mathbb{R}^d \times [0, 1]$, let $0 < \alpha \leq \beta$ and let $(f, g)$ be a Fenchel-Legendre pair such that $g' : \mathbb{R} \to \mathbb{R}$ is $[\alpha, \beta]$ bi-Lipschitz. Suppose that for a predictor $p : \mathbb{R}^d \to \mathrm{ran}(g')$ we have*

$$\mathcal{L}_g(f' \circ p \,; D) \leq \min_{\|\mathbf{w}\|_2 \leq B} \mathcal{L}_g(\mathbf{w} \,; D) + \varepsilon \tag{2.1}$$

*Then we also have:* $\mathrm{err}_2(p) \leq \frac{\beta}{\alpha} \cdot \mathrm{opt}(\mathrm{GLM}_{g', B}) + 2\beta\varepsilon$.

The proof of Theorem 2.1 is based on the following pointwise distortion bound between matching losses corresponding to bi-Lipschitz link functions and the squared distance.

**Lemma 2.2** (Pointwise Distortion Bound for bi-Lipschitz link functions). *Let $0 < \alpha \leq \beta$ and let $(f, g)$ be a Fenchel-Legendre pair such that $f' : \mathrm{ran}(g') \to \mathbb{R}$ is $[\frac{1}{\beta}, \frac{1}{\alpha}]$ bi-Lipschitz. Then for any $y, p \in \mathrm{ran}(g')$ we have*

$$\ell_g(y, f'(p)) - \ell_g(y, f'(y)) = \mathbb{D}_f(y, p) \in \left[ \frac{1}{2\beta}(y - p)^2, \frac{1}{2\alpha}(y - p)^2 \right]$$

In the case that $f'$ is differentiable on $(0, 1)$, the proof of Lemma 2.2 follows from an application of Taylor's approximation theorem of degree 2 on the function $f$, since the Bregman divergence $\mathbb{D}_f(y, p)$ is exactly equal to the error of the second degree Taylor's approximation of $f(y)$ around $p$ and $f''(\xi) \in [\frac{1}{\beta}, \frac{1}{\alpha}]$ for any $\xi \in \operatorname{ran}(g')$. The relationship between $\ell_g$ and $\mathbb{D}_f$ follows from property (1.2). Note that when $g'$ is $[\alpha, \beta]$ bi-Lipschitz, then $f'$ is $[\frac{1}{\beta}, \frac{1}{\alpha}]$ bi-Lipschitz.

Theorem 2.1 follows by applying Lemma 2.2 appropriately to bound the error of a predictor $p$ by its matching loss $\mathcal{L}_g(f' \circ p)$ and bound the matching loss of the linear function corresponding to $\mathbf{w}^*$ by the squared error of $g'(\mathbf{w}^* \cdot \mathbf{x})$, where $g'(\mathbf{w}^* \cdot \mathbf{x})$ is the element of $\operatorname{GLM}_{g', B}$ with minimum squared error.

Although Theorem 2.1 only applies to bi-Lipschitz activations $g'$, it has the advantage that the assumption it makes on $p$ corresponds to a convex optimization problem and, when the marginal distribution has certain concentration properties (for generalization), can be solved efficiently through gradient descent on the empirical loss function. As a consequence, for bi-Lipschitz activations we can obtain $O(\mathrm{opt})$ efficiently under mild distributional assumptions in the agnostic setting.

## 3   Agnostically Learning Single-Index Models

In this section, we provide our main result on agnostically learning SIMs. We combine the distortion bounds we established in Section 2 with results from Gopalan et al. [2023] on Omniprediction, which can be used to learn a predictor $p$ that satisfies the guarantee of Theorem 2.1 simultaneously for all bi-Lipschitz activations. By doing so, we obtain a result for all Lipschitz and non-decreasing activations simultaneously.

**Theorem 3.1** (Agnostically Learning SIMs). *Let $\lambda, B \geq 1$, $\varepsilon > 0$ and let $D$ be a distribution over $\mathbb{R}^d \times [0, 1]$ with second moments bounded by $\lambda$. Then, Algorithm 1 agnostically learns the class $\operatorname{SIM}_B$ over $D$ up to squared error $O(B\sqrt{\lambda}\sqrt{\operatorname{opt}(\operatorname{SIM}_B, D)}) + \varepsilon$ using time and sample complexity $\operatorname{poly}(d, B, \lambda, \frac{1}{\varepsilon})$. Moreover, the same is true for any algorithm with an omniprediction guarantee like the one of Theorem 3.2.*

In order to apply Theorem 2.1, we use the following theorem which is a combination of results in Gopalan et al. [2023], where they show that the matching losses corresponding to a wide class of functions can all be minimized simultaneously by an efficiently computable predictor.

**Theorem 3.2** (Omnipredictors for Matching Losses, combination of results in Gopalan et al. [2023])**.** *Let $\lambda, L, R, B \geq 1$, $\varepsilon > 0$ and let $D$ be a distribution over $\mathbb{R}^d \times [0, 1]$ whose marginal on $\mathbb{R}^d$ has $\lambda$-bounded second moments. Then, Algorithm 1, given sample access to $D$, with probability at least $2/3$ returns a predictor $p : \mathbb{R}^d \to (0, 1)$ with the following guarantee. For any Fenchel-Legendre pair $(f, g)$ such that $g' : \mathbb{R} \to \mathbb{R}$ is $L$-Lipschitz, and $f'$ takes values within the interval $[-R, R]$, $p$ satisfies*

$$\mathcal{L}_g(f' \circ p \, ; D) \leq \min_{\|\mathbf{w}\|_2 \leq B} \mathcal{L}_g(\mathbf{w} \, ; D) + \varepsilon.$$

*The algorithm requires time and sample complexity $\operatorname{poly}(\lambda, B, L, R, \frac{1}{\varepsilon})$.*

Algorithm 1 is a version of the algorithm of Gopalan et al. [2023] for calibrated multiaccuracy, specific to our setting. For a more quantitative version of Theorem 3.2, see Theorem C.3 in the appendix.

We aim to apply Theorem 3.2 to the class of all Lipschitz activations (which is wider than the class of bi-Lipschitz activations). This is enabled by the following lemma, whose proof is based on Theorem 2.1 and the fact that the error of a predictor is bounded by the sum of the error of another predictor and the squared expected distance between the two predictors.

**Lemma 3.3.** *Let $D$ be a distribution over $\mathbb{R}^d \times [0, 1]$. Let $g' : \mathbb{R} \to \mathbb{R}$ be some fixed activation, and $f'$ its dual. Consider the class $\operatorname{GLM}_{g', B}$, and let $\mathbf{w}^*$ be the weights achieving $\operatorname{opt}(\operatorname{GLM}_{g', B}, D)$. Let $\phi' : \mathbb{R} \to \mathbb{R}$ be an $[\alpha, \beta]$ bi-Lipschitz function (differentiable except possibly at finitely many points) that we wish to approximate $g'$ by. Any predictor $p : \mathbb{R}^d \to \mathbb{R}$ that satisfies*

$$\mathcal{L}_\phi(f' \circ p \, ; D) \leq \min_{\|\mathbf{w}\|_2 \leq B} \mathcal{L}_\phi(\mathbf{w} \, ; D) + \varepsilon$$

*also satisfies the following $\ell_2$ error guarantee:*

$$\operatorname{err}_2(p) \leq \frac{2\beta}{\alpha} \operatorname{opt}(\operatorname{GLM}_{g', B}) + \frac{2\beta}{\alpha} \mathbb{E}\left[\left(g'(\mathbf{w}^* \cdot \mathbf{x}) - \phi'(\mathbf{w}^* \cdot \mathbf{x})\right)^2\right] + 2\beta\varepsilon.$$

---

**Algorithm 1:** Calibrated Multiaccuracy (modification of Algorithm 2 in Gopalan et al. [2023])

---

**Input:** Sufficiently large training set $S$ over $\mathbb{R}^d \times [0,1]$, parameters $\varepsilon$, $\lambda$, $B$, $L$, $R$

**Output:** A predictor $p : \mathbb{R}^d \to (0,1)$

Form a training set $S'$ by substituting each element $(\mathbf{x}, y)$ in $S$ with $(\mathbf{x}, y')$ where $y' \in \{0,1\}$ is formed as follows. Conditioned on $y$, let $y'$ be an independent Bernoulli random variable with probability of success $y$.

Let $A = B \cdot L$, $C > 0$ a sufficiently large universal constant and let $\delta = \frac{\varepsilon^2}{CA^2\lambda R^2}$, $\sigma = \frac{\varepsilon - \delta R}{2CA^2\lambda R}$.
Set $p \leftarrow 0$ (the zero function).

**repeat**

> **repeat**
>
> > Let $S''$ be a subset of $S'$ with $|S''| = C\frac{d^2 A^2 \lambda R^2}{\varepsilon^2}$ and set $S' \leftarrow S' \setminus S''$.
> > Form a set $T$ by substituting each element $(\mathbf{x}, y')$ of $S''$ with $(\mathbf{x}, z)$, where
> > $z = y' - p(\mathbf{x})$.
> > Let $\mathbf{v} = \frac{1}{|T|} \sum_{(\mathbf{x},z) \in T} z\mathbf{x}$.
> > **Reject** if $\|\mathbf{v}\|_2 \leq 3(\varepsilon - A\sqrt{\lambda\delta})/(4A)$.
> > **Otherwise**, let $\mathbf{w} = B\mathbf{v}/\|\mathbf{v}\|_2$ and update $p$ to be the following predictor
> >
> > $$\mathbf{x} \mapsto (p(\mathbf{x}) + \sigma(\mathbf{w} \cdot \mathbf{x}))_{[0,1]}, \text{ where } (t)_{[0,1]} = \min\{\max\{t, 0\}, 1\}$$
>
> **until** *some iteration rejects (which happens when it has found a multiaccurate predictor)*;
>
> Let $p^\delta : \mathbb{R}^d \to [0,1]$ such that $p^\delta(\mathbf{x}) = \sum_{j=0}^{1/\delta} j\delta \mathbb{1}\{p(\mathbf{x}) \in I_j\}$, where
> $I_j = [(j - \frac{1}{2})\delta, (j + \frac{1}{2})\delta]$.
> Let $S''$ be a subset of $S'$ with $|S''| = C\frac{A^8\lambda^4 R^8}{\varepsilon^8} \log(\frac{A\lambda R}{\varepsilon})$, set $S' \leftarrow S' \setminus S''$ and let
> $S''_j = \{(\mathbf{x}, y') \in S'' : p^\delta(\mathbf{x}) = j\delta\}$.
> Estimate the calibration error $|\mathbb{E}_{p^\delta(\mathbf{x})} \mathbb{E}_{y|p^\delta(\mathbf{x})}[y - p^\delta(\mathbf{x})]|$ of $p^\delta$ using the samples in $S''$
> according to the following expression
>
> $$\sum_{j=0}^{1/\delta} \frac{|S''_j|}{|S''|} |j\delta - \bar{y}_j|, \text{ where } \bar{y}_j = \frac{1}{|S''_j|} \sum_{(\mathbf{x},y') \in S''_j} y'$$
>
> If the estimate has value at most $\varepsilon/2$, then **output** $p^\delta$ and **terminate**.
> **Otherwise**, update $p$ to be the following (calibrated) predictor
>
> $$\mathbf{x} \mapsto \sum_{j=0}^{1/\delta} \bar{y}_j \mathbb{1}\{p(\mathbf{x}) \in I_j\}$$

**until** *some iteration terminates the execution*;

---

By combining Theorem 3.2 with Lemma 3.3 (whose proofs can be found in Appendix C), we are now ready to prove our main theorem.

*Proof of Theorem 3.1.* We will combine Theorem 3.2, which states that there is an efficient algorithm that simultaneously minimizes the matching losses corresponding to bounded, non-decreasing and Lipschitz activations, with Lemma 3.3, which implies that minimizing the matching loss corresponding to the nearest bi-Lipschitz activation is sufficient to obtain small error. Note that we may assume that $\varepsilon < 1/2$, since otherwise the problem is trivial (output the zero function and pick $C = 2$).

As a first step, we show that link functions corresponding to bi-Lipschitz activations are bounded (according to Definition C.1, for $\gamma = 0$). In particular, let $\phi' : \mathbb{R} \to \mathbb{R}$ be an $[\alpha, \beta]$ bi-Lipschitz activation for some $\beta \geq \alpha > 0$ such that $\phi'(s) \in [-1, 2]$ for some $s \in \mathbb{R}$ and let $\psi'$ be the inverse of $\phi'$ ($\phi'$ is invertible since it is strictly increasing). We will show that $\psi'(r) \in [-R, R]$ for any $r \in [0, 1]$ and $R = O(|s| + 1/\alpha)$.

We pick $r_0 = 0$, $r_1 = 1$ and get that $|\psi'(\phi'(s)) - \psi'(r_0)| \leq \frac{1}{\alpha}|\phi'(s) - r_0| \leq \frac{2}{\alpha}$. Hence $\psi'(r_0) \geq \psi'(\phi'(s)) - \frac{2}{\alpha} = s - \frac{1}{\alpha}$. Similarly, we get $\psi'(r_1) \leq s + \frac{2}{\alpha}$. Therefore, $\psi'(r) \in [\psi'(0), \psi'(1)] \subseteq [-|s| - \frac{2}{\alpha}, |s| + \frac{2}{\alpha}]$, for any $r \in [0, 1]$, due to monotonicity of $\psi'$.

For a given non-decreasing and 1-Lipschitz $g'$, we will now show that there is a bounded bi-Lipschitz activation $\phi'$ such that if the assumption of Lemma 3.3 is satisfied for $\phi'$ by a predictor $p$, then the error of $p$ is bounded by

$$\mathrm{err}_2(p) \leq O\big(B\sqrt{\lambda}(\mathrm{opt}(\mathrm{GLM}_{g',B}))^{1/2}\big) + O(\lambda B^2 \varepsilon)$$

Suppose, first, that $\mathrm{opt}(\mathrm{GLM}_{g',B}) \leq \varepsilon^2$. Then, we pick $\phi'(t) = g'(t) + \varepsilon t$. Note that $\phi'$ is $[\varepsilon, 1 + \varepsilon]$ bi-Lipschitz. Moreover, since $\mathrm{opt}(\mathrm{GLM}_{g',B}) \leq \varepsilon^2$, we must have some $s \in \mathbb{R}$ with $|s| \leq 2\lambda B^2$ such that $g'(s) \in [-1, 2]$. Otherwise, we would have $\mathrm{opt}(\mathrm{GLM}_{g',B}) = \mathbf{E}[(g'(\mathbf{w}^* \cdot \mathbf{x}) - y)^2] \geq \mathbf{E}[(g'(\mathbf{w}^* \cdot \mathbf{x}) - y)^2 \mid |\mathbf{w}^* \cdot \mathbf{x} \leq 2\lambda B^2|] \cdot \mathbb{P}[|\mathbf{w}^* \cdot \mathbf{x}| \leq 2\lambda B^2] \geq \mathbb{P}[|\mathbf{w}^* \cdot \mathbf{x}| \leq 2\lambda B^2] = 1 - \mathbb{P}[|\mathbf{w}^* \cdot \mathbf{x}| > 2\lambda B^2] \geq \frac{1}{4} > \varepsilon^2$, due to Chebyshev's inequality, the fact that $\mathbf{w}^* \in \mathcal{W}$ and the bounded moments assumption. Therefore, $\psi'$ is $(R = 2\lambda B^2 + \frac{2}{\varepsilon}, \gamma = 0)$-bounded and we have

$$\mathbb{E}\left[\left(g'(\mathbf{w}^* \cdot \mathbf{x}) - \phi'(\mathbf{w}^* \cdot \mathbf{x})\right)^2\right] \leq \varepsilon^2 \, \mathbb{E}[(\mathbf{w}^* \cdot \mathbf{x})^2] \leq \varepsilon^2 \lambda B^2$$

As a consequence, under the assumption of Lemma 3.3 for $\phi'$, the error of the corresponding predictor $p$ is $\mathrm{err}_2(p) \leq 2(1 + \varepsilon)\varepsilon + 2(1 + \varepsilon)\varepsilon\lambda B^2 + 2(1 + \varepsilon)\varepsilon = O(\lambda B^2 \varepsilon)$.

In the case that $\mathrm{opt}(\mathrm{GLM}_{g',B}) > \varepsilon^2$, we pick $\phi'(t) = g'(t) + t\frac{\sqrt{\mathrm{opt}(\mathrm{GLM}_{g',B})}}{B\sqrt{\lambda}}$. We may also assume that $\mathrm{opt}(\mathrm{GLM}_{g',B}) \leq 1/2$, since otherwise any predictor with range $[0, 1]$ will have error at most $2\mathrm{opt}(\mathrm{GLM}_{g',B})$. Then, $\psi$ is $(O(\lambda B^2 + \frac{B\sqrt{\lambda}}{\varepsilon}), 0)$-bounded, $\phi'$ is $[\frac{1}{B}\sqrt{\mathrm{opt}(\mathrm{GLM}_{g',B})}/\sqrt{\lambda}, 1 + \frac{1}{B}]$ bi-Lipschitz which gives

$$\mathbb{E}\left[\left(g'(\mathbf{w}^* \cdot \mathbf{x}) - \phi'(\mathbf{w}^* \cdot \mathbf{x})\right)^2\right] \leq \frac{\mathrm{opt}(\mathrm{GLM}_{g',B})}{B^2\lambda} \, \mathbb{E}[(\mathbf{w}^* \cdot \mathbf{x})^2] \leq \mathrm{opt}(\mathrm{GLM}_{g',B})$$

As a consequence, under the assumption of Lemma 3.3 for $\phi'$, the error of the corresponding predictor $p$ is $\mathrm{err}_2(p) \leq 4(1 + \frac{1}{B})B\sqrt{\lambda}\sqrt{\mathrm{opt}(\mathrm{GLM}_{g',B})} + 2(1 + \frac{1}{B})\varepsilon$. Using a similar approach as for the case $\mathrm{opt}(\mathrm{GLM}_{g',B}) \leq \varepsilon$, we can show that $\psi'$ is polynomially bounded (as per Definition C.1), since $\mathrm{opt}(\mathrm{GLM}_{g',B}) \leq \frac{1}{2}$.

To conclude the proof of Theorem 3.1, we apply Theorem 3.2 with appropriate (polynomial) choice of parameters $(R = O(\lambda B^2 + \frac{B\sqrt{\lambda}}{\varepsilon}), L = 2)$, to show that there is an efficient algorithm that outputs a predictor $p : \mathbb{R}^d \to (0, 1)$ for which the assumption of Lemma 3.3 holds simultaneously for all bi-Lipschitz activations ($\phi'$) with sufficiently bounded inverses ($\psi'$). $\qquad\square$

## 4 Stronger Guarantees for Logistic Regression

In this section, we follow the same recipe we used in Section 2 to get distortion bounds similar to Theorem 2.1 for the sigmoid activation (or, equivalently, the logistic model) under the assumption that the marginal distribution is sufficiently concentrated (see Definition C.1). In particular, Theorem 2.1 does not hold, since the sigmoid is not bi-Lipschitz and our main Theorem 3.1 only provides a guarantee of $O(\sqrt{\mathrm{opt}})$ for squared error. We use appropriate pointwise distortion bounds for the matching loss corresponding to the sigmoid activation and provide guarantees of $\widetilde{O}(\mathrm{opt})$ for logistic regression with respect to both squared and absolute error, under appropriate assumptions about the concentration of the marginal distribution. The proofs of this section are provided in Appendix D.

For the logistic model, the link function $f'$ is defined as $f'(r) = \ln(\frac{r}{1-r})$, for $r \in (0, 1)$ and the corresponding activation $g'$ is the sigmoid $g'(t) = \frac{1}{1+e^{-t}}$ for $t \in \mathbb{R}$. The corresponding matching loss is the logistic loss.

**Squared error.** We first provide a result for squared loss minimization. In comparison to Theorem 2.1, the qualitative interpretation of the following theorem is that, while the sigmoid activation is not bi-Lipschitz, it is effectively bi-Lipschitz under sufficiently concentrated marginals.

**Theorem 4.1** (Squared Loss Minimization through Logistic Loss Minimization). *Let $D$ be a distribution over $\mathbb{R}^d \times [0,1]$ whose marginal on $\mathbb{R}^d$ is $(1,2)$-concentrated. Let $g' : \mathbb{R} \to \mathbb{R}$ be the sigmoid activation, i.e., $g'(t) = (1 + e^{-t})^{-1}$ for $t \in \mathbb{R}$. Assume that for some $B > 0$, $\varepsilon > 0$ and a predictor $p : \mathbb{R}^d \to (0,1)$ we have*

$$\mathcal{L}_g(f' \circ p \,; D) \leq \min_{\mathbf{w}:\|\mathbf{w}\|_2 \leq B} \mathcal{L}_g(\mathbf{w} \,; D) + \varepsilon \tag{4.1}$$

*If we let $\mathrm{opt}_g = \min_{\|\mathbf{w}\|_2 \leq B} \mathrm{err}_2(g'_\mathbf{w})$, then for the predictor $p$ and some universal constant $C > 0$ we also have*

$$\mathrm{err}_2(p) \leq C \, \mathrm{opt}_g \exp\left( B^2 + \sqrt{B^2 \, \log \frac{1}{\mathrm{opt}_g}} \right) + 2\varepsilon.$$

In particular, the squared error of $p$ is upper bounded by $\widetilde{O}(\mathrm{opt}_g)$, since the function $t \mapsto \exp(\log^{1/2} t)$ is asymptotically smaller than any polynomial function $t \mapsto t^\gamma$ with $\gamma > 0$.

Once more, the proof of our result is based on an appropriate pointwise distortion bound which we provide below. It follows by the fact that the Bregman divergence corresponding to the sigmoid is the Kullback-Leibler divergence and by combining Pinsker's inequality (lower bound) with Lemma 4.1 of Götze et al. [2019] (upper bound).

**Lemma 4.2** (Pointwise Distortion Bound for Sigmoid). *Let $g'$ be the sigmoid activation. Then, for any $y, p \in (0,1)$ we have that $\mathbb{D}_f(y,p) = \mathbb{D}_{\mathrm{KL}}(y\|p) = y \ln(y/p) + (1-y)\ln(\frac{1-y}{1-p})$. Moreover*

$$\ell_g(y, f'(p)) - \ell_g(y, f'(y)) = \mathbb{D}_{\mathrm{KL}}(y\|p) \in \left[ \frac{1}{2}(y-p)^2, \ \frac{2}{\min\{p, 1-p\}} \cdot (y-p)^2 \right]$$

We translate Lemma 4.2 to a bound on the squared error of a matching loss minimizer (in this case, the logistic loss) using an approach similar to the one for Theorem 2.1. In order to use the upper bound on the surrogate loss provided by Lemma 4.2, we apply it to $p \leftarrow g'(\mathbf{w} \cdot \mathbf{x})$, where $g'$ is the sigmoid function, and observe that the quantity $\frac{1}{p(1-p)}$ is exponential in $|\mathbf{w} \cdot \mathbf{x}|$. Hence, when the marginal is $(\lambda, 2)$-concentrated (subgaussian concentration), then $\frac{1}{p(1-p)}$ is effectively bounded.

**Absolute error.** All of the results we have provided so far have focused on squared error minimization. We now show that our approach yields results even for the absolute error, which can also be viewed as learning in the p-concept model [Kearns and Schapire, 1994]. In particular, for a distribution $D$ over $\mathbb{R}^d \times [0,1]$, we define the absolute error of a predictor $p : \mathbb{R}^d \to [0,1]$ as follows.

$$\mathrm{err}_1(p) = \mathop{\mathbb{E}}_{(\mathbf{x},y)\sim D}[|y - p(\mathbf{x})|]$$

In the specific case when the labels are binary, i.e., $y \in \{0,1\}$, we have

$$\mathrm{err}_1(p) = \mathop{\mathbb{E}}_{(\mathbf{x},y)\sim D}[|y - p(\mathbf{x})|] = \mathop{\mathbb{P}}_{(\mathbf{x},y,y_p)\sim D_p}[y \neq y_p] \qquad \text{(see Proposition A.2)}$$

where the distribution $D_p$ is over $\mathbb{R}^d \times \{0,1\} \times \{0,1\}$ and is formed by drawing samples $(\mathbf{x}, y)$ from $D$ and, given $\mathbf{x}$, forming $y_p$ by drawing a conditionally independent Bernoulli random variable with parameter $p(\mathbf{x})$. We provide the following result.

**Theorem 4.3** (Absolute Loss Minimization through Logistic Loss Minimization). *Let $D$ be a distribution over $\mathbb{R}^d \times \{0,1\}$ whose marginal on $\mathbb{R}^d$ is $(1,1)$-concentrated. Let $g' : \mathbb{R} \to \mathbb{R}$ be the sigmoid activation, i.e., $g'(t) = (1 + e^{-t})^{-1}$ for $t \in \mathbb{R}$. Assume that for some $B > 0$, $\varepsilon > 0$ and a predictor $p : \mathbb{R}^d \to (0,1)$ we have*

$$\mathcal{L}_g(f' \circ p \,; D) \leq \min_{\mathbf{w}:\|\mathbf{w}\|_2 \leq B} \mathcal{L}_g(\mathbf{w} \,; D) + \varepsilon \tag{4.2}$$

*If we let $\mathrm{opt}_g = \min_{\|\mathbf{w}\|_2 \leq B} \mathrm{err}_1(g'_\mathbf{w})$, then for the predictor $p$ and some universal constant $C > 0$ we also have*

$$\mathrm{err}_1(p) \leq C \, B \, \mathrm{opt}_g \, \log \frac{1}{\mathrm{opt}_g} + \varepsilon$$

The corresponding distortion bound in this case is between the absolute and logistic losses and works when the labels are binary.

**Lemma 4.4** (Pointwise Distortion between Absolute and Logistic Loss). *Let $g'$ be the sigmoid activation. Then, there is a constant $c \in \mathbb{R}$ such that for any $y \in \{0, 1\}$ and $p \in (0, 1)$, we have*

$$\ell_g(y, f'(p)) - c \in \left[ |y - p| \, , \, 2 \cdot \ln\left(\frac{1}{p(1-p)}\right) \cdot |y - p| \right]$$

The bound of Theorem 4.3 implies an algorithm for learning an unknown sigmoid neuron in the p-concept model, by minimizing a convex loss. While there are algorithms achieving stronger guarantees [Diakonikolas et al., 2022b] for agnostically learning sigmoid neurons, such algorithms typically make strong distributional assumptions including concentration, anti-concentration and anti-anti-concentration or boundedness.

Moreover, it is useful to compare the bound we provide in Theorem 4.3 to a lower bound by Diakonikolas et al. [2020c, Theorem 4.1], which concerns the problem of agnostically learning halfspaces by minimizing convex surrogates. In particular, they show that even under log-concave marginals, no convex surrogate loss can achieve a guarantee better than $O(\mathrm{opt}\log(1/\mathrm{opt}))$, where opt is measured with respect to the $\ell_1$ error (which is equal to the probability of error). The result is not directly comparable to our upper bound, since we examine the sigmoid activation. Their setting can be viewed as a limit case of ours by letting the norm of the weight vector grow indefinitely (the sigmoid tends to the step function), but the main complication is that our upper bound is of the form $O(B\mathrm{opt}\log(1/\mathrm{opt}))$, which scales with $B$. However, their lower bound concerns marginal distributions that are not only concentrated, but are also anti-concentrated and anti-anticoncentrated, while our results only make concentration assumptions.

## 5 Necessity of Norm Dependence

In this final section, we use a lower bound due to Diakonikolas et al. [2022a] on agnostic learning of GLMs using SQ algorithms and compare it with our main result (Theorem 3.1). For simplicity, we specialize to the case of the standard sigmoid or logistic function. A modification of their proof ensures that the bound holds under isotropic marginals.[2]

**Theorem 5.1** (SQ Lower Bound for Agnostically Learning GLMs, variant of [Diakonikolas et al., 2022a, Thm C.3]). *Let $g' : \mathbb{R}^d \to \mathbb{R}$ be the standard logistic function. Any SQ algorithm either requires $d^{\omega(1)}$ queries or $d^{-\omega(1)}$ tolerance to distinguish between the following two labeled distributions:*

- *(Labels have signal.) $D_{\mathrm{signal}}$ on $\mathbb{R}^d \times \mathbb{R}$ is such that $\mathrm{opt}(\mathrm{GLM}_{g',B}, D_{\mathrm{signal}}) \leq \exp(-\Omega(\log^{1/4} d)) = o(1)$ for some $B = \mathrm{poly}(d)$.*

- *(Labels are random.) $D_{\mathrm{random}}$ on $\mathbb{R}^d \times \mathbb{R}$ is such that the labels $y$ are drawn i.i.d. from $\{a, b\}$ for certain universal constants $a, b \in [0, 1]$. In particular, $\mathrm{opt}(\mathrm{GLM}_{g',B}, D_{\mathrm{random}}) = \Omega(1)$ for any $B$.*

*Both $D_{\mathrm{signal}}$ and $D_{\mathrm{random}}$ have the same marginal on $\mathbb{R}^d$, with 1-bounded second moments.*

Let us consider applying our main theorem (Theorem 3.1) to this setting, with $D$ being either $D_{\mathrm{signal}}$ or $D_{\mathrm{random}}$, and with the same $B = \mathrm{poly}(d)$ as is required to achieve small error in the "labels have signal" case. We would obtain a predictor with $\ell_2$ error at most $B\sqrt{\mathrm{opt}(\mathrm{GLM}_{g',B})}$ (or indeed with $\mathrm{SIM}_B$ in place of $\mathrm{GLM}_{g',B}$). Since this is $\omega(1)$, this guarantee is insufficient to distinguish the two cases above, which is as it should be since our main algorithm indeed fits into the SQ framework.

Theorem 5.1 does, however, justify a dependence on the norm $B$ in our main result. In particular, it is clear that a guarantee of the form $\mathrm{opt}(\mathrm{GLM}_{g',B})^c$ for any universal constant $c > 0$ (independent of $B$) would be too strong, as it would let us distinguish the two cases above. In fact, this lower bound rules out a large space of potential error guarantees stated as functions of $B$ and $\mathrm{opt}(\mathrm{GLM}_{g',B})$. For instance, for sufficiently large $d$, it rules out any error guarantee of the form $\exp(O(\log^{1/5} B)) \cdot \mathrm{opt}(\mathrm{GLM}_{g',B})^{c'}$ for any universal constant $c' > 0$.

---

[2]Specifically, our features correspond to all multilinear monomials (or parities) of degree at most $k$ over $\{\pm 1\}^n$, whereas they use all monomials (not necessarily multilinear) of degree at most $k$. These yield equivalent representations since the hard distributions are obtained from the uniform distribution on $\{\pm 1\}^n$.

## Acknowledgments and Disclosure of Funding

We wish to thank the anonymous reviewers of NeurIPS 2023 for their constructive feedback. Aravind Gollakota was at UT Austin while this work was done, supported by NSF award AF-1909204 and the NSF AI Institute for Foundations of Machine Learning (IFML). Adam R. Klivans was supported by NSF award AF-1909204 and the NSF AI Institute for Foundations of Machine Learning (IFML). Konstantinos Stavropoulos was supported by NSF award AF-1909204, the NSF AI Institute for Foundations of Machine Learning (IFML), and by scholarships from Bodossaki Foundation and Leventis Foundation.

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

# A Technical Lemmas

In this section, we provide some technical Lemmas that we use in our proofs.

**Proposition A.1** (Weak Learner for Linear Functions). *Let $D$ be a distribution over $\mathbb{R}^d \times [-1, 1]$ whose marginal on $\mathbb{R}^d$ has $\lambda$-bounded second moments and $B > 0$. For any $\varepsilon > 0$ and $\delta \in (0, 1)$, there is a universal constant $C > 0$ and an algorithm that given a set $S$ of i.i.d. samples from $D$ of size at least $C \cdot \frac{d^2 \lambda B^2}{\varepsilon^2} \log \frac{1}{\delta}$, runs in time $O(d |S|)$ and satisfies the following specifications with probability at least $1 - \delta$*

1. *If $\mathbb{E}_{(\mathbf{x}, z) \sim D}[z(\mathbf{w} \cdot \mathbf{x})] \geq \varepsilon$ for some $\mathbf{w} \in \mathbb{R}^d$ with $\|\mathbf{w}\|_2 \leq B$, then the algorithm accepts. Otherwise, it may or may not reject and return a special symbol.*

2. *If the algorithm accepts then it returns $\mathbf{w} \in \mathbb{R}^d$ with $\|\mathbf{w}\|_2 \leq B$ such that we have $\mathbb{E}_{(\mathbf{x}, z) \sim D}[z(\mathbf{w} \cdot \mathbf{x})] \geq \varepsilon/4$.*

*Proof.* We will prove the proposition for $\delta = 1/6$. We may boost the probability of success with repetition.

The algorithm computes the vector $\mathbf{v} = \mathbb{E}_S[z\, \mathbf{x}]$. If $\|\mathbf{v}\|_2 \leq \frac{3\varepsilon}{4B}$, then the algorithm rejects and outputs a special symbol. Otherwise, it outputs the vector $\frac{B}{\|\mathbf{v}\|_2} \mathbf{v}$.

Suppose, first, that $\mathbb{E}_{(\mathbf{x}, z) \sim D}[z(\mathbf{w} \cdot \mathbf{x})] \geq \varepsilon$ for some $\mathbf{w}$ with $\|\mathbf{w}\|_2 \leq B$. Then, due to Chebyshev's inequality we have for any $i \in [d]$

$$\mathbb{P}\left[\left|\mathbb{E}_S[z\,\mathbf{x}_i] - \mathbb{E}_D[z\,\mathbf{x}_i]\right| > \frac{\varepsilon}{8\,B\sqrt{d}}\right] \leq \frac{64\, d\, B^2}{|S|\,\varepsilon^2}\, \mathbb{E}[\mathbf{x}_i^2] \leq \frac{64\, d\, B^2\, \lambda}{|S|\,\varepsilon^2} \leq \frac{1}{6\,d} \quad \text{(for large enough } |S|)$$

Hence, with probability at least $5/6$, we have $\|\mathbb{E}_S[z\mathbf{x}] - \mathbb{E}_D[z\mathbf{x}]\|_2 \leq \frac{\varepsilon}{8B}$, due to a union bound. Therefore, $\|\mathbf{v}\|_2 \geq \|\mathbb{E}_D[z\mathbf{x}]\|_2 - \frac{\varepsilon}{8B} \geq \frac{\mathbb{E}_D[z(\mathbf{w}\cdot\mathbf{x})]}{B} - \frac{\varepsilon}{8B} \geq \frac{7B}{8}$ and the algorithm accepts.

Suppose, now, that the algorithm accepts. Then, we have $\|\mathbf{v}\|_2 > \frac{3\varepsilon}{4B}$ and (with probability at least $5/6$) we have

$$\mathbb{E}_D\left[\frac{B}{\|\mathbf{v}\|_2} z(\mathbf{v} \cdot \mathbf{x})\right] = \frac{B}{\|\mathbf{v}\|_2} \mathbf{v} \cdot \mathbb{E}[z\mathbf{x}] \geq \varepsilon/4$$

since $\|\mathbb{E}_S[z\mathbf{x}] - \mathbb{E}_D[z\mathbf{x}]\|_2 \leq \frac{\varepsilon}{8B}$. This concludes the proof. $\qquad\square$

**Proposition A.2.** *Let $D$ be a distribution over $\mathbb{R}^d \times \{0, 1\}$ and $p : \mathbb{R}^d \to [0, 1]$. Consider the distribution $D_p$ over $\mathbb{R}^d \times \{0, 1\} \times \{0, 1\}$, which is formed by drawing samples $(\mathbf{x}, y)$ from $D$ and, given $\mathbf{x}$, forming $y_p$ by drawing a conditionally independent Bernoulli random variable with parameter $p(\mathbf{x})$. Then we have*

$$\mathrm{err}_1(p) = \mathop{\mathbb{E}}_{(\mathbf{x}, y) \sim D}[|y - p(\mathbf{x})|] = \mathop{\mathbb{P}}_{(\mathbf{x}, y, y_p) \sim D_p}[y \neq y_p]$$

*Proof.* Since over $D_p$, $y$ and $y_p$ are conditionally independendent, we have

$$\begin{aligned}
\mathrm{err}_1(p) = \mathbb{E}[|y - p(\mathbf{x})|] &= \mathbb{E}[(1 - p(\mathbf{x}))\mathbb{1}\{y = 1\} + p(\mathbf{x})\mathbb{1}\{y = 0\}] \\
&= \mathop{\mathbb{E}}_{\mathbf{x}}\Big[\mathbb{P}[y_p = 0|\mathbf{x}]\,\mathbb{P}[y = 1|\mathbf{x}] + \mathbb{P}[y_p = 1|\mathbf{x}]\,\mathbb{P}[y = 0|\mathbf{x}]\Big] \\
&= \mathop{\mathbb{E}}_{x}[\mathbb{P}[y \neq y_p|\mathbf{x}]] = \mathbb{P}[y \neq y_p]
\end{aligned}$$

$\qquad\square$

# B Proofs from Section 2

## B.1 Proof of Theorem 2.1

To prove Theorem 2.1, we first prove the following more general theorem. Theorem 2.1 may then be easily recovered from this by setting $\mathcal{H} = \mathrm{GLM}_{g', B}$ and observing that $f'(g'(\mathbf{w} \cdot \mathbf{x})) = \mathbf{w} \cdot \mathbf{x}$, since $g'$ is invertible.

**Theorem B.1** (Squared Error Minimization through Distorted Matching Loss Minimization). *Let $D$ be a distribution over $\mathbb{R}^d \times [0,1]$, let $0 < \alpha \le \beta$ and let $(f, g)$ be a pair of Fenchel-Legendre dual functions such that $g' : \mathbb{R} \to \mathbb{R}$ is continuous, non-decreasing and $f' : \mathrm{ran}(g') \to \mathbb{R}$ is $[\frac{1}{\beta}, \frac{1}{\alpha}]$ bi-Lipschitz. Let $\varepsilon > 0$ and $\mathcal{H} \subseteq \{\mathbb{R}^d \to \mathrm{ran}(g')\}$. Assume that for a predictor $p : \mathbb{R}^d \to \mathrm{ran}(g')$ we have*

$$\mathcal{L}_g(f' \circ p \,; D) \le \min_{h \in \mathcal{H}} \mathcal{L}_g(f' \circ h \,; D) + \varepsilon \tag{B.1}$$

*Then, for the predictor $p$, we also have:* $\mathrm{err}_2(p) \le \frac{\beta}{\alpha} \cdot \min_{h \in \mathcal{H}} \mathrm{err}_2(h) + 2\beta\varepsilon$.

*Proof.* We apply Lemma 2.2 with $y \leftarrow y$ and $p \leftarrow p(\mathbf{x})$ and take expectations over $D$ on both sides. We have that

$$\mathrm{err}_2(p) \le 2\beta \cdot \mathbb{E}\,\ell_g(y, f'(p(\mathbf{x}))) - 2\beta \cdot \mathbb{E}\,\ell_g(y, f'(y))$$

Therefore, we can bound the squared error of $p$ as follows.

$$\begin{aligned}
\mathrm{err}_2(p) &\le 2\beta \cdot \mathcal{L}_g(f' \circ p \,; D) - 2\beta \cdot Q^* \\
&\le 2\beta \cdot \mathcal{L}_g(f' \circ h \,; D) - 2\beta \cdot Q^* \qquad\qquad \text{(by assumption, for any } h \in \mathcal{H})
\end{aligned}$$

where $Q^* = \mathbb{E}\,\ell_g(y, f'(y))$.

We now apply Lemma 2.2 again with $y \leftarrow y$ and $p \leftarrow h(\mathbf{x})$ and we similarly have

$$\mathbb{E}\,\ell_g(y, f' \circ h(\mathbf{x})) - Q^* \le \frac{1}{2\alpha}\mathrm{err}_2(h)$$

Therefore, for any $h \in \mathcal{H}$, we have, in total: $\mathrm{err}_2(p) \le \frac{\beta}{\alpha}\mathrm{err}_2(h) + 2\beta\varepsilon$. $\qquad\square$

We first prove Lemma 2.2, which we restate here for convenience.

**Lemma B.2.** *Assume $f'$ is $[1/\beta, 1/\alpha]$ bi-Lipschitz and differentiable on all except from a finite number of points on any bounded interval. Then for any $y, p \in \mathrm{ran}(g')$ we have*

$$\ell_g(y, f'(p)) - \ell_g(y, f'(y)) = D_f(y, p) \in \left[ \frac{1}{2\beta}(y - p)^2, \frac{1}{2\alpha}(y - p)^2 \right]$$

*Proof.* We first show that $\ell_g(y, f'(p)) - \ell_g(y, f'(y)) = D_f(y, p)$. In particular, we have

$$g(f'(p)) = f'(p)g'(f'(p)) - f(g'(f'(p))) = pf'(p) - f(p),$$

since $f'(p) \in \mathrm{ran}(f')$ and we know that $g(t) = tg'(t) - f(g'(t))$ for any $t \in \mathrm{ran}(f')$ as well as $g'(f'(p)) = p$ for any $p \in \mathrm{ran}(g')$. Therefore, we have

$$\begin{aligned}
\ell_g(y, f'(p)) - \ell_g(y, f'(y)) &= g(f'(p)) - yf'(p) - g(f'(y)) + yf'(y) \\
&= pf'(p) - f(p) - yf'(p) - yf'(y) + f(y) + yf'(y) = \\
&= f(y) - f(p) - (y - p)f'(p) = D_f(y, p).
\end{aligned}$$

Let $\psi : \mathrm{ran}(g') \to \mathbb{R}$ be such that $\psi'(p) = f'(p)$ and $\psi'$ is differentiable on the open interval between $y$ and $p$, with $\psi''(\xi) \in [1/\beta, 1/\alpha]$ for any $\xi$ between $y$ and $p$. Let $\gamma_y := f(y) - \psi(y), \gamma_p := f(p) - \psi(p)$ and $\gamma_\psi := 2\max\{|\gamma_y|, |\gamma_p|\}$. Then we have that

$$D_f(y, p) = \psi(y) - \psi(p) - (y - p)\psi'(p) + (\gamma_y - \gamma_p) = \frac{1}{2}\psi''(\xi)(y - p)^2 + (\gamma_y - \gamma_p)$$

$$D_f(y, p) \in \left[ \frac{1}{2\beta}(y - p)^2 - 2\gamma_\psi, \frac{1}{2\alpha}(y - p)^2 + 2\gamma_\psi \right],$$

for any $\psi$ as defined above (say $\psi \in \Psi$). In particular, we have

$$D_f(y, p) \le \frac{1}{2\alpha}(y - p)^2 + 2\inf_{\psi \in \Psi} \gamma_\psi \text{ and}$$

$$D_f(y, p) \ge \frac{1}{2\beta}(y - p)^2 - 2\inf_{\psi \in \Psi} \gamma_\psi$$

Since, we have only a finite number of points where the derivative is not well defined, a simple smoothening technique may give us $\Psi$ such that $\inf_{\psi \in \Psi} \gamma_\psi = 0$. $\qquad\square$

# C  Proofs from Section 3

## C.1  Proof of Theorem 3.2

We first define a boundedness property which we use in order to apply the results from Gopalan et al. [2023]. The property states that the activation function (the partial inverse of the link function) must either have a range that covers all possible labels, or has a range whose closure covers all possible labels and the rate with which the labels are covered as we tend to the limits of the domain is at least polynomial. For example, the sigmoid activation tends to 1 (resp. 0) exponentially fast as its argument increases (resp. decreases).

**Definition C.1** (Bounded Functions). Let $u : (0,1) \to \mathbb{R}$ be a non-decreasing function defined on the interval $(0,1)$. For $R, \gamma \geq 0$, we say that $u$ is $(R, \gamma)$-bounded on $[0,1]$ if for any $\varepsilon > 0$, there are $r_0 \leq r_1 \in [0,1]$ such that if we let $u(r_i) = \lim_{r \to r_i} u(r)$, $i = 0, 1$ then

$$\max\{-u(r_0), u(r_1)\} \leq R \left(\frac{1}{\varepsilon}\right)^{\gamma}$$
$$(1 - r_1)(u(r) - u(r_1)) \leq \varepsilon \text{ for } r \geq r_1 \text{ and}$$
$$r_0(u(r_0) - u(r)) \leq \varepsilon \text{ for } r \leq r_0$$

**Proposition C.2.** *Let $u : (0,1) \to (-R, R)$ be non-decreasing and continuous, then $u$ is $(R, 0)$-bounded.*

We restate a more quantitative version of Theorem 3.2 here for convenience.

**Theorem C.3** (Omnipredictors for Matching Losses, combination of results in Gopalan et al. [2023])**.** *Let $D$ be a distribution over $\mathbb{R}^d \times [0,1]$ whose marginal on $\mathbb{R}^d$ has $\lambda$-bounded second moments. There is an algorithm that, given sample access to $D$, with high probability returns a predictor $p : \mathbb{R} \to (0,1)$ with the following guarantee. For any pair of Fenchel-Legendre dual functions $(f, g)$ such that $g' : \mathbb{R} \to \mathbb{R}$ is continuous, non-decreasing and $L$-Lipschitz, and $f'$ is $(R, \gamma)$-bounded (see Definition C.1), $p$ satisfies*

$$\mathcal{L}_g(f' \circ p \,;\, D) \leq \min_{\|\mathbf{w}\|_2 \leq B} \mathcal{L}_g(\mathbf{w} \,;\, D) + \varepsilon.$$

*The algorithm requires time*

$$O \left( d^3 B^4 L^4 \lambda^2 R^3 \left(\frac{3}{\varepsilon}\right)^{3+3\gamma} + dB^2 L^2 \lambda R^4 \left(\frac{3}{\varepsilon}\right)^{4+4\gamma} + B^{10} L^{10} \lambda^5 R^{12} \left(\frac{3}{\varepsilon}\right)^{12+12\gamma} \log \left(\frac{BL\lambda R}{\varepsilon^{1+\gamma}}\right) \right)$$

*and sample complexity*

$$O \left( d^2 B^4 L^4 \lambda^2 R^3 \left(\frac{3}{\varepsilon}\right)^{3+3\gamma} + B^8 L^8 \lambda^4 R^{10} \left(\frac{3}{\varepsilon}\right)^{10+10\gamma} \log \left(\frac{BL\lambda R}{\varepsilon^{1+\gamma}}\right) \right)$$

*Proof of Theorem 3.2.* The idea of the proof (given by Gopalan et al. [2023]) is that in each repetition of both the inner and the outer loop of Algorithm 1, there is a potential function which reduces by some amount that is bounded away above zero. The potential function is in fact the function $\mathbf{E}_D[(p^*(\mathbf{x}) - p(\mathbf{x}))^2]$, where $p^*(\mathbf{x}) = \mathbf{E}_D[y'|\mathbf{x}]$ (for us, $y'$ is formed by drawing a Bernoulli random variable with probability of success $y$, given an example $(\mathbf{x}, y) \in \mathbb{R}^d \times [0,1]$ drawn from $D$). Since $\mathbf{E}_D[(p^*(\mathbf{x}) - p(\mathbf{x}))^2] \in [0,1]$, the number of iterations of each of the loops has to be bounded. Moreover, after the completion of each of the inner loops, the current value of $p$ corresponds to an approximately multiaccurate predictor and note that the algorithm terminates if and only if a discretized version $p^\delta$ of (the multiaccurate) $p$ is approximately calibrated. The output is then $p^\delta$ which is close to $p$ (and hence also approximately multiaccurate), but also approximately calibrated. We will now present a small number of technical modifications we need to make in the proof of Gopalan et al. [2023] in order to specialize it to our setting. The main difference here is that we do not consider the distribution of $\mathbf{x}$ to have bounded norm with probability 1, but we only assume it to have bounded second moments. For the following, we assume that $g$ is 1-Lipschitz, since we can set $B \leftarrow B \cdot L$ and push the Lipschitz constant in the domain of $\mathbf{w}$.

Suppose first that the marginal of $D$ on $\mathbb{R}^d$ is supported on the unit ball $\mathbb{B}_d$ and that the labels are binary. Then, the result would follow from Theorems 7.7 and A.4 of Gopalan et al. [2023]. In particular,

Theorem 7.7 states that given access to a weak learner with the specifications of Proposition A.1, there is an efficient algorithm that computes an $\varepsilon_1$-calibrated and $(\mathcal{C}, \varepsilon_1)$-multiaccurate predictor $p$, where the notions of calibration and multiaccuracy originate to the literature of fairness and are defined, e.g., in Definitions 3.1 and 3.2 of Gopalan et al. [2023] and $\mathcal{C} = \{\mathbf{x} \to \mathbf{w} \cdot \mathbf{x} \mid \|\mathbf{w}\|_2 \leq B\} \cup \{\mathbf{x} \to 1\}$ (the class $\mathcal{C}$ is bounded as long as $\|\mathbf{x}\|_2 \leq 1$ almost surely). Theorem A.4 states that for $\varepsilon_2 > 0$, any $\varepsilon_1$-calibrated and $(\mathcal{C}, \varepsilon_1)$-multiaccurate predictor $p$ minimizes simultaneously the matching loss corresponding to any pair $(f, g) \in \mathcal{F}$ (where $f$ is $(R, \gamma)$-bounded) up to error

$$R(1/\varepsilon_2)^\gamma \varepsilon_1 + \varepsilon_1 + \varepsilon_2$$

The expression above is formed by proving that any pair $(f, g) \in \mathcal{F}$ has the property that $f'$ is $(\varepsilon_2, R(1/\varepsilon_2)^\gamma)$-approximately optimal (as per the Definition A.1 of Gopalan et al. [2023]), for any $\varepsilon_2 > 0$. In particular, we would like to show that for any $\varepsilon_2 > 0$ there exists $\widehat{f'}$ such that the following is true for any $r \in [0, 1]$

$$\ell_g(r, \widehat{f'}(r)) \leq \ell_g(r, f'(r)) + \varepsilon_2$$
$$|\widehat{f'}(r)| \leq R \cdot (1/\varepsilon_2)^\gamma$$

We may pick $\widehat{f'}$ as follows (for $r_0 \leq r_1$ as given by Definition C.1 for $\varepsilon \leftarrow \varepsilon_2$, since $f'$ is $(R, \gamma)$-bounded).

$$\widehat{f'}(r) = \begin{cases} f'(r), & \text{if } r \in [r_0, r_1] \\ f'(r_0), & \text{if } r < r_0 \\ f'(r_1), & \text{if } r > r_1 \end{cases}$$

The desired result follows from using the expression for $\ell_g$, the convexity of $g$ (since $g'$ is non decreasing) and the guarantees of Definition C.1. In particular, we have that $|\widehat{f'}| \leq \max\{|f'(r_0)|, |f'(r_1)|\}$ since $f'$ is increasing. Moreover, for $r \in [r_0, r_1]$ we have $\ell_g(r, \widehat{f'}(r)) = \ell_g(r, f'(r))$ and for $r < r_0$ we have

$$\begin{aligned}
\ell_g(r, \widehat{f'}(r)) = \ell_g(r, f'(r_0)) &= g(f'(r_0)) - rf'(r_0) \\
&\leq r_0(f'(r_0) - f'(r)) + g(f'(r)) - rf'(r_0) && \text{(since } g \text{ is convex)} \\
&\leq r_0(f'(r_0) - f'(r)) + g(f'(r)) - rf'(r) && \text{(since } f' \text{ is increasing and } r < r_0) \\
&\leq \varepsilon_2 + g(f'(r)) - rf'(r) && \text{(since } f' \text{ is bounded according to Definition C.1)} \\
&= \ell_g(r, f'(r)) + \varepsilon_2
\end{aligned}$$

Similarly, for $r > r_1$, we have

$$\begin{aligned}
\ell_g(r, \widehat{f'}(r)) = \ell_g(r, f'(r_1)) &= g(f'(r_1)) - rf'(r_1) \\
&\leq r_1(f'(r_1) - f'(r)) + g(f'(r)) - rf'(r_1) && \text{(since } g \text{ is convex)} \\
&\leq (1 - r_1)(f'(r_1) - f'(r)) + g(f'(r)) + (1 - r)f'(r_1) - f'(r) \\
&\leq (1 - r_1)(f'(r_1) - f'(r)) + g(f'(r)) - rf'(r) && \text{(since } f' \text{ is increasing, } 1 > r > r_1) \\
&\leq \varepsilon_2 + g(f'(r)) - rf'(r) && \text{(since } f' \text{ is bounded according to Definition C.1)} \\
&= \ell_g(r, f'(r)) + \varepsilon_2
\end{aligned}$$

In order to acquire $R(1/\varepsilon_2)^\gamma \varepsilon_1 + \varepsilon_1 + \varepsilon_2 \leq \varepsilon$, we set $\varepsilon_2 = \varepsilon/3$ and $\varepsilon_1 = (\varepsilon/3)^{1+\gamma}/R$.

However, we only assume that the marginal distribution has $\lambda$-bounded second moments and we, therefore, need to make certain modifications to the proof of their Theorem 7.7. In particular, the boundedness assumption is used in the proofs of Lemma 7.2, Lemma 7.6 and Theorem 7.7 in Gopalan et al. [2023].

During the execution of Algorithm 1, two types of updates are made. The first type of update is done within the inner loop and corresponds to beginning from a predictor $p_{\text{old}}$ and acquiring $p_{\text{new}}$ which is the function $\mathbf{x} \mapsto (p_{\text{old}}(\mathbf{x}) + \sigma(\mathbf{w} \cdot \mathbf{x}))_{[0,1]}$, where $\mathbf{w}$ is the output of the weak learner of our Proposition A.1, run with $\varepsilon \leftarrow \varepsilon_3$ ($\varepsilon_3$ will be specified later). To lower bound the decrease in the potential function during this type of update, we use a version of Lemma 7.6 in Gopalan et al. [2023]. In this case, one needs to pick a step size $\sigma$ that is polynomially smaller than the guarantee that the

weak learner provides. In particular, if the weak learner of our Proposition A.1 is run with $\varepsilon \leftarrow \varepsilon_3$, then one acquires (following the proof of Lemma 7.6 in Gopalan et al. [2023])

$$\mathbf{E}[(p^*(\mathbf{x}) - p_{\mathrm{old}}(\mathbf{x}))^2] - \mathbf{E}[(p^*(\mathbf{x}) - p_{\mathrm{new}}(\mathbf{x}))^2] \geq \frac{\sigma \cdot \varepsilon_3}{2} - B^2 \lambda \sigma^2 .$$

If $\sigma$ is picked to be $\sigma = \frac{\varepsilon_3}{4B^2\lambda}$, then the quantity of interest $\mathbf{E}[(p^*(\mathbf{x}) - p_{\mathrm{old}}(\mathbf{x}))^2] - \mathbf{E}[(p^*(\mathbf{x}) - p_{\mathrm{new}}(\mathbf{x}))^2]$ is lower bounded by $\frac{\varepsilon_3^2}{16B^2\lambda}$. Hence the number of iterations of the inner loop is upper bounded by $O(\frac{B^2\lambda}{\varepsilon_3^2})$. Note that in their original algorithm, $\sigma$ was picked equal to $\varepsilon_3$ and this is why $\varepsilon_3$ (or another corresponding parameter) does not appear in their proofs. The updated choice of $\sigma$ generates a polynomial overhead in time and sample complexity. We pick $\varepsilon_3 = \frac{1}{2}(\varepsilon_1 - B\sqrt{\lambda\delta})$ so that each time we exit the inner loop, we have that, with high probability, the current value of $p$ corresponds to an $(\varepsilon_1 - B\sqrt{\lambda\delta})$-multiaccurate predictor.

The second type of update (the outer loop update) is a calibration step where $p_{\mathrm{new}}$ is set to be the following function with the notation of Algorithm 1

$$\mathbf{x} \mapsto \sum_{j=0}^{1/\delta} \bar{y}_j \mathbb{1}\{p_{\mathrm{old}}(\mathbf{x}) \in I_j\}$$

In this case, Corollary 7.5 of Gopalan et al. [2023] can be used as is to acquire that $\mathbf{E}[(p^*(\mathbf{x}) - p_{\mathrm{old}}(\mathbf{x}))^2] - \mathbf{E}[(p^*(\mathbf{x}) - p_{\mathrm{new}}(\mathbf{x}))^2] \geq \varepsilon_1^2/8$ if we set $\delta \leq \varepsilon_1^2/C$ for some large enough universal constant $C > 0$. Hence, the number of iterations of the outer loop is upper bounded by $O(1/\varepsilon_1^2)$.

It remains to show that once the algorithm terminates, the output is approximately calibrated and multiaccurate. Regarding multiaccuracy, whenever we exit the inner loop, the current value of $p$ corresponds (with high probability) to an $(\varepsilon_1 - B\sqrt{\lambda\delta})$-multiaccurate predictor. The predictor $p^\delta$ is close to $p$ in absolute distance and therefore can be shown to be approximately multiaccurate using some version of Lemma 7.2 in Gopalan et al. [2023]. In particular, following the proof of Lemma 7.2, the multiaccuracy guarantee for $p^\delta$ changes to $(\mathcal{C}, \alpha + B\sqrt{\lambda\delta})$-multiaccuracy, where $\alpha$ is the multiaccuracy guarantee for $p$, by using a Cauchy-Schwarz inequality and bounding $\mathbb{E}[(\mathbf{w} \cdot \mathbf{x})^2]$ by $B^2 \cdot \lambda$. Since $\alpha = \varepsilon_1 - B\sqrt{\lambda\delta}$, the multiaccuracy guarantee is $\varepsilon_1$. The calibration guarantee follows by the fact that the termination criterion corresponds to empirically checking whether $p^\delta$ is calibrated and by using Lemma 7.4 in Gopalan et al. [2023], the empirical estimate should be close to the true calibration error. Overall, once we terminate, the output is $\varepsilon_1$ calibrated and multiaccurate.

Since the time and sample complexity of each of the calls of the weak learner depends on $\varepsilon_3$, we pick $\delta \leftarrow \frac{\varepsilon_1^2}{C(1+B^2\lambda)}$ so that $\varepsilon_3 = \Theta(\varepsilon_1)$ and $\delta \leq \frac{\varepsilon_1^2}{C}$ as required.

**Sample Complexity.** For each of the inner loop iterations, we require a fresh sample of size $O(\frac{d^2 B^2 \lambda}{\varepsilon_1^2})$, as specified by Proposition A.1 (by setting the probability of success to a sufficiently small constant). For the outer loop, we require $O(\frac{1+B^8\lambda^4}{\varepsilon_1^8} \log(\frac{1+B\lambda}{\varepsilon_1}))$ samples per iteration, so that the calibration error estimate is accurate enough, according to Lemma 7.4 in Gopalan et al. [2023].

**Time Complexity.** Each of the inner loop iterations requires $O(\frac{d^3 B^2 \lambda}{\varepsilon_1^2})$ time and each of the outer loop iterations requires an additional $O(d\frac{1+B^2\lambda}{\varepsilon_1^2} + \frac{1+B^{10}\lambda^5}{\varepsilon_1^{10}} \log(\frac{1+B\lambda}{\varepsilon_1}))$ time.

The final technical complication we need to address is that their algorithm works only given binary labels. We can, however, form binary labels as follows. Let $(\mathbf{x}, y)$ be drawn from $D$. We have that $y \in [0, 1]$. Given $y$, we draw an independent Bernoulli random variable $y'$ with probability of success $y$, forming the distribution $D'$ over $\mathbb{R}^d \times \{0, 1\}$. We run the algorithm of Gopalan et al. [2023] on $D'$ and receive a predictor $p$ such that

$$\mathcal{L}_g(f' \circ p \,; D') \leq \min_{\mathbf{w} \in \mathcal{W}} \mathcal{L}_g(\mathbf{w} \,; D') + \varepsilon, \ \text{ for any } (f, g) \in \mathcal{F}$$

We have that

$$
\begin{aligned}
\mathcal{L}_g(c\,;D') &= \mathop{\mathbb{E}}_{\mathbf{x},y'}\left[g(c(\mathbf{x})) - y'c(\mathbf{x})\right]\\
&= \mathop{\mathbb{E}}_{\mathbf{x}}\left[g(c(\mathbf{x})) - \mathop{\mathbb{E}}_{y}\left[\mathop{\mathbb{E}}_{y'}\left[y'\,\middle|\,y\right]\,\middle|\,\mathbf{x}\right]c(\mathbf{x})\right]\\
&= \mathop{\mathbb{E}}_{\mathbf{x}}\left[g(c(\mathbf{x})) - \mathop{\mathbb{E}}_{y}\left[y\,\middle|\,\mathbf{x}\right]c(\mathbf{x})\right]\\
&= \mathop{\mathbb{E}}_{\mathbf{x},y}\left[g(c(\mathbf{x})) - yc(\mathbf{x})\right] = \mathcal{L}_g(c\,;D)
\end{aligned}
$$

This concludes the proof of Theorem 3.2. $\qquad\square$

## C.2 Proof of Lemma 3.3

We first apply Theorem 2.1 with $g' \leftarrow \phi'$ to get that for $\phi'_{\mathbf{w}}(\mathbf{x}) = \phi'(\mathbf{w}\cdot\mathbf{x})$, we have

$$
\mathrm{err}_2(p) \le \frac{\beta}{\alpha}\,\mathrm{err}_2(\phi'_{\mathbf{w}^*}) + 2\beta\varepsilon \qquad\qquad \text{(since inequality holds for } \mathbf{w}\in\mathcal{W})
$$

Moreover, we have

$$
\begin{aligned}
\mathrm{err}_2(\phi'_{\mathbf{w}^*}) &= \mathbb{E}\left[\left(y - \phi'(\mathbf{w}^*\cdot\mathbf{x})\right)^2\right]\\
&= \mathbb{E}\left[\left(y - g'(\mathbf{w}^*\cdot\mathbf{x}) + g'(\mathbf{w}^*\cdot\mathbf{x}) - \phi'(\mathbf{w}^*\cdot\mathbf{x})\right)^2\right]\\
&\le 2\,\mathbb{E}\left[\left(y - g'(\mathbf{w}^*\cdot\mathbf{x})\right)^2\right] + 2\,\mathbb{E}\left[\left(g'(\mathbf{w}^*\cdot\mathbf{x}) - \phi'(\mathbf{w}^*\cdot\mathbf{x})\right)^2\right]\\
&= 2\,\mathrm{opt}_g + 2\,\mathbb{E}\left[\left(g'(\mathbf{w}^*\cdot\mathbf{x}) - \phi'(\mathbf{w}^*\cdot\mathbf{x})\right)^2\right]
\end{aligned}
$$

This concludes the proof of lemma 3.3.

# D  Proofs from Section 4

## D.1 Proof of Theorem 4.1

In the case we consider here, $g'$ is the sigmoid activation, i.e., $g'(t) = (1 + e^{-t})^{-1}$ for any $t \in \mathbb{R}$. In particular, the pointwise surrogate loss we consider satisfies

$$
\ell_g(y, f'(p)) = y\ln\frac{1}{p} + (1-y)\ln\frac{1}{1-p} - \ln 2\,,
$$

for any $y \in [0,1]$ and $p \in (0,1)$. We may extend Lemma 4.2 to also capture $y \in \{0,1\}$, by defining $\ell_g(0, f'(0)) = \ell_g(1, f'(1)) = -\ln 2$ (the inequality would hold under this definition). Hence, following a similar procedure as the one used for proving Theorem B.1, we obtain the following by applying Lemma 4.2

$$
\mathrm{err}_2(p) \le 2\left(\mathcal{L}_g(f'\circ p) - \mathbb{E}[\ell_g(y, f'(y))]\right) \tag{D.1}
$$

$$
\mathcal{L}_g(\mathbf{w}^*) - \mathbb{E}[\ell_g(y, f'(y))] \le \mathbb{E}\left[\frac{2}{g'(\mathbf{w}^*\cdot\mathbf{x})\vee(1 - g'(\mathbf{w}^*\cdot\mathbf{x}))}\cdot(y - g'(\mathbf{w}^*\cdot\mathbf{x}))^2\right] \tag{D.2}
$$

$$
\mathcal{L}_g(f'\circ p) \le \mathcal{L}_g(\mathbf{w}^*) + \varepsilon \tag{D.3}
$$

Therefore, in order to prove Theorem 4.1, it is sufficient to provide a strong enough upper bound for the quantity of the right hand side of Equation (D.2) in terms of $\mathrm{opt}_g$. We observe that

$$
\frac{2}{g'(\mathbf{w}^*\cdot\mathbf{x})\vee(1 - g'(\mathbf{w}^*\cdot\mathbf{x}))} \le 4\exp(|\mathbf{w}^*\cdot\mathbf{x}|)\,, \text{ for any } \mathbf{x}\in\mathbb{R}^d
$$

It remains to bound the quantity $\mathbb{E}[e^{|\mathbf{w}^* \cdot \mathbf{x}|}(y - g'_{\mathbf{w}}(\mathbf{x}))^2]$. Set $Q = e^{|\mathbf{w}^* \cdot \mathbf{x}|}(y - g'_{\mathbf{w}}(\mathbf{x}))^2$ ($Q$ is a random variable). Then for any $r \geq 0$ we have

$$
\begin{aligned}
\mathbb{E}[Q] &= \mathbb{E}[Q \cdot \mathbb{1}\{|\mathbf{w}^* \cdot \mathbf{x}| \leq r\}] + \mathbb{E}[Q \cdot \mathbb{1}\{|\mathbf{w}^* \cdot \mathbf{x}| > r\}] \\
&\leq e^r \, \mathbb{E}[(y - g'_{\mathbf{w}^*}(\mathbf{x}))^2 \mathbb{1}\{|\mathbf{w}^* \cdot \mathbf{x}| \leq r\}] + \mathbb{E}[e^{|\mathbf{w}^* \cdot \mathbf{x}|} \mathbb{1}\{|\mathbf{w}^* \cdot \mathbf{x}| > r\}] \\
&\leq e^r \cdot \mathrm{opt} + \mathbb{E}[e^{|\mathbf{w}^* \cdot \mathbf{x}|} \mathbb{1}\{|\mathbf{w}^* \cdot \mathbf{x}| > r\}]
\end{aligned}
$$

To bound the quantity $\mathbb{E}[e^{|\mathbf{w}^* \cdot \mathbf{x}|} \mathbb{1}\{|\mathbf{w}^* \cdot \mathbf{x}| > r\}]$, consider $F(s) := \Pr[e^{|\mathbf{w}^* \cdot \mathbf{x}|} \mathbb{1}\{|\mathbf{w}^* \cdot \mathbf{x}| > r\} \geq s]$. We have that

$$
F(s) = \mathbb{1}\{s = 0\}\Pr[|\mathbf{w}^* \cdot \mathbf{x}| \leq r] + \Pr[|\mathbf{w}^* \cdot \mathbf{x}| \geq \max\{\ln s, r\}] \, .
$$

Since $\mathbb{E}[e^{|\mathbf{w}^* \cdot \mathbf{x}|} \mathbb{1}\{|\mathbf{w}^* \cdot \mathbf{x}| > r\}] = \int_{s=0}^{\infty} F(s) \, ds = \int_{s=0}^{\infty} \Pr[|\mathbf{w}^* \cdot \mathbf{x}| \geq \max\{\ln s, r\}] \, ds$, we obtain

$$
\begin{aligned}
\mathbb{E}[Q] &= \int_{s=0}^{\infty} \Pr\left[|\mathbf{w}^* \cdot \mathbf{x}| \geq \max\{\ln s, r\}\right] ds \\
&\leq \int_{s=0}^{e^r} \Pr\left[|\widehat{\mathbf{w}}^* \cdot \mathbf{x}| \geq \frac{r}{B}\right] ds + \int_{s=e^r}^{\infty} \Pr\left[|\widehat{\mathbf{w}}^* \cdot \mathbf{x}| \geq \frac{\ln s}{B}\right] ds && \text{(since } \|\mathbf{w}^*\|_2 \leq B) \\
&\leq e^r \cdot e^{-(r/B)^2} + \int_{s=e^r}^{\infty} e^{-(\ln s / B)^2} ds && \text{(see Def. 1.6)} \\
&= e^r \cdot e^{-(r/B)^2} + e^r \cdot \int_{u=0}^{\infty} e^{u - \left(\frac{u+r}{B}\right)^2} du && \text{(define } u = \ln s - r) \\
&\leq e^r \cdot e^{-(r/B)^2} + e^r \cdot e^{-(r/B)^2} \cdot B \cdot e^{\frac{B^2}{2}} \cdot \int_{u=0}^{\infty} e^{-(u - \frac{B}{2})^2} du \\
&\leq C e^{B^2} e^r e^{-(r/B)^2}
\end{aligned}
$$

Therefore, in total, we have that $\mathrm{err}_2(p) \leq 8 e^r \mathrm{opt}_g + 8 C e^{B^2} e^r e^{-(r/B)^2} + 2\varepsilon$ and we may obtain Theorem 4.1 by picking $r = B(\ln \frac{1}{\mathrm{opt}})^{1/2}$.

## D.2 Proof of Theorem 4.3

We first prove Lemma 4.4. We have that for any $y \in \{0, 1\}$ and $p \in (0, 1)$

$$
\ell_g(y, f'(p)) = y \ln \frac{1}{p} + (1 - y) \ln \frac{1}{1 - p} - \ln 2 = \mathrm{CE}(y, p) - \ln 2 \, ,
$$

where $\mathrm{CE}(y, p)$ is the cross entropy function. It is sufficient to show that for $y \in \{0, 1\}$ and $p \in [0, 1]$,

$$
|y - p| \leq \mathrm{CE}(y, p) \leq 2|y - p| \log\left(\frac{1}{p(1 - p)}\right) \tag{D.4}
$$

Observe that

$$
\mathrm{CE}(0, p) = \log\left(\frac{1}{1 - p}\right) = \sum_{i=1}^{\infty} \frac{p^i}{i} \tag{D.5}
$$

where the series on the right converges for all $p < 1$. We can also write

$$
\mathrm{CE}(1, p) = \log\left(\frac{1}{p}\right) = \sum_{i=1}^{\infty} \frac{(1 - p)^i}{i} \tag{D.6}
$$

with the series converging for $p > 0$.

For the lower bound, we observe the following inequalities hold for all $p \in [0, 1]$

$$
\begin{aligned}
\mathrm{CE}(0, p) &\geq p = |0 - p| \\
\mathrm{CE}(1, p) &\geq 1 - p = |1 - p|.
\end{aligned}
$$

For the upper bound, we first prove the claim for $y = 0$, where it states that

$$\mathrm{CE}(0, p) = \log\left(\frac{1}{1-p}\right) \leq 2p \log\left(\frac{1}{p(1-p)}\right). \tag{D.7}$$

When $p \leq 1/2$ we can use Eq. (D.5) to bound

$$\mathrm{CE}(0, p) \leq \sum_{i=1}^{\infty} \frac{p^i}{i} \leq \sum_{i=1}^{\infty} p^i = \frac{p}{1-p} \leq 2p. \tag{D.8}$$

The bounds holds by observing that since $p(1-p) \geq 1/4$,

$$\log\left(\frac{1}{p(1-p)}\right) \geq \log(4) \geq 1$$

When $p \geq 1/2$, we note that

$$\log\left(\frac{1}{p(1-p)}\right) \geq \log\left(\frac{1}{1-p}\right) = \mathrm{CE}(0, p)$$

and $2p \geq 1$. Hence the bound holds in this case too.

In the case where $y = 1$, the bound states that

$$\mathrm{CE}(1, p) = \log\left(\frac{1}{p}\right) \leq 2(1-p) \log\left(\frac{1}{p(1-p)}\right).$$

This is implied by our bound for $y = 0$ by taking $q = 1 - p$. This concludes the proof of Lemma 4.4 and we are ready to prove Theorem 4.3. The following are true

$$\mathrm{err}_1(p) \leq \mathcal{L}_g(f' \circ p) - \ln 2 \tag{D.9}$$

$$\mathcal{L}_g(\mathbf{w}^*) - \ln 2 \leq 2 \cdot \mathbb{E}\left[\ln\left(\frac{1}{g'(\mathbf{w}^* \cdot \mathbf{x}) \cdot (1 - g'(\mathbf{w}^* \cdot \mathbf{x}))}\right) \cdot |y - g'(\mathbf{w}^* \cdot \mathbf{x})|\right] \tag{D.10}$$

$$\mathcal{L}_g(f' \circ p) \leq \mathcal{L}_g(\mathbf{w}^*) + \varepsilon \tag{D.11}$$

Similarly to the proof of Theorem 4.1, we observe that

$$\ln\left(\frac{1}{g'(\mathbf{w}^* \cdot \mathbf{x}) \cdot (1 - g'(\mathbf{w}^* \cdot \mathbf{x}))}\right) \leq \ln 4 + |\mathbf{w}^* \cdot \mathbf{x}|$$

It remains to bound the quantity $\mathbb{E}[|\mathbf{w}^* \cdot \mathbf{x}| \cdot |y - g'_{\mathbf{w}}(\mathbf{x})|]$. Set $Q = |\mathbf{w}^* \cdot \mathbf{x}| \cdot |y - g'_{\mathbf{w}}(\mathbf{x})|$ ($Q$ is a random variable). Then for any $r \geq 0$ we have

$$\mathbb{E}[Q] = \mathbb{E}[Q \cdot \mathbb{1}\{|\mathbf{w}^* \cdot \mathbf{x}| \leq r\}] + \mathbb{E}[Q \cdot \mathbb{1}\{|\mathbf{w}^* \cdot \mathbf{x}| > r\}]$$
$$\leq r\, \mathbb{E}[|y - g'_{\mathbf{w}^*}(\mathbf{x})| \cdot \mathbb{1}\{|\mathbf{w}^* \cdot \mathbf{x}| \leq r\}] + \mathbb{E}[|\mathbf{w}^* \cdot \mathbf{x}| \cdot \mathbb{1}\{|\mathbf{w}^* \cdot \mathbf{x}| > r\}]$$
$$\leq r \cdot \mathrm{opt} + \mathbb{E}[|\mathbf{w}^* \cdot \mathbf{x}| \cdot \mathbb{1}\{|\mathbf{w}^* \cdot \mathbf{x}| > r\}]$$

To bound the quantity $\mathbb{E}[|\mathbf{w}^* \cdot \mathbf{x}| \cdot \mathbb{1}\{|\mathbf{w}^* \cdot \mathbf{x}| > r\}]$, consider $F(s) := \Pr[|\mathbf{w}^* \cdot \mathbf{x}| \cdot \mathbb{1}\{|\mathbf{w}^* \cdot \mathbf{x}| > r\} \geq s]$. We have that

$$F(s) = \mathbb{1}\{s = 0\} \Pr[|\mathbf{w}^* \cdot \mathbf{x}| \leq r] + \Pr[|\mathbf{w}^* \cdot \mathbf{x}| \geq \max\{s, r\}].$$

Since $\mathbb{E}[|\mathbf{w}^* \cdot \mathbf{x}| \cdot \mathbb{1}\{|\mathbf{w}^* \cdot \mathbf{x}| > r\}] = \int_{s=0}^{\infty} F(s)\, ds = \int_{s=0}^{\infty} \Pr[|\mathbf{w}^* \cdot \mathbf{x}| \geq \max\{s, r\}]\, ds$, we obtain

$$\mathbb{E}[Q] = \int_{s=0}^{\infty} \Pr[|\mathbf{w}^* \cdot \mathbf{x}| \geq \max\{s, r\}]\, ds$$
$$\leq \int_{s=0}^{r} \Pr\left[|\widehat{\mathbf{w}}^* \cdot \mathbf{x}| \geq \frac{r}{B}\right] ds + \int_{s=r}^{\infty} \Pr\left[|\widehat{\mathbf{w}}^* \cdot \mathbf{x}| \geq \frac{s}{B}\right] ds \qquad (\text{since } \|\mathbf{w}^*\|_2 \leq B)$$
$$\leq r \cdot e^{-r/B} + \int_{s=r}^{\infty} e^{-s/B}\, ds \qquad (\text{see Def. } 1.6)$$
$$= r \cdot e^{-r/B} + B \cdot e^{-r/B}$$

Therefore, in total, we have that $\mathrm{err}_1(p) \leq 2 \ln 4 \cdot \mathrm{opt}_g + 2(r + B)e^{-r/B} + 2r \cdot \mathrm{opt}_g + \varepsilon$ and we may obtain Theorem 4.3 by picking $r = B \cdot \ln\frac{1}{\mathrm{opt}}$.

