# OpenReview forum: "Agnostically Learning Single-Index Models using Omnipredictors"
_NeurIPS.cc/2023/Conference — NeurIPS 2023 poster_

### Official Review · Reviewer_uXX7 · 2023-07-06

**Soundness:** 3 good
**Presentation:** 1 poor
**Contribution:** 3 good
**Rating:** 6
**Confidence:** 4

**Summary:**

The paper studies the problem of learning SIMs agnostically. The authors proposed an algorithm achieving $B\sqrt{opt}$ $\ell_2$-error under mild distributional assumptions. Their main contributions are twofold: 1) they linked the $\ell_2$ loss of a bi-lipschitz activation with its matching loss. 2) they proposed an algorithm that minimizes the matching loss for all Lipschitz activations. They also showed stronger guarantees for logistic regression and argued that the dependence of norm $B$ is inevitable in the final $\ell_2$ loss bound achievable for any efficient algorithm.

**Strengths:**

The authors bring in the technique of fenchel duality to study the distortion bound between the $\ell_2$ loss and the matching loss for GLMs. Even though their idea is similar to theorem 5.5 in [GHK+23], the result is original. In addition, the authors proposed a method using omnipredictors to find a predictor that minimizes the matching loss with respect to all 1-Lipschitz activations. This idea is new to the reviewer.

**Weaknesses:**

The result about the $\ell_2$ error of learning sigmoid activation (thm 4.1) seems weak, as there are already similar results in the literature, see for example, thm 3.3 [FCG20], though in [FCG20] they used lightly stronger assumptions (bounded $D_x$), their $\ell_2$ error does not have the logarithm factor. A detailed comparison with related works would be helpful to prove the significance of this paper.

The presentation of this paper lacks clarity and is confusing. There is no clear description of the algorithm proposed by the authors, and the sample complexity and runtime are not clearly stated throughout the paper. There are also a lot of symbols used in the paper that are not defined, to name a few: the term $opt_g$ appeared in the proof of thm 3.1, the $\varepsilon$ appeared in the statement of thm C.2, and the function $l_2$ used in the proof of thm 3.2.

The proofs lack readability. For example, in line 261, it is hard to understand why $opt_g\geq Pr[|w^*\cdot x|\leq 2\lambda B^2]$ without further clarification. It would also be helpful to have more details in the proof of thm 3.2, like the relation between $\epsilon_1$ and $\epsilon_2$, and the statement from line 533 to 537.


**Questions:**

Can the authors provide some intuition about multi-accuracy and multi-caliberation? How do they relate to the omnipredictors for SIMs and matching loss? In particular, what exactly is the relation between multi-accuracy and the gradient of the matching loss, as mentioned in line 110?

**Limitations:**

I agree with the authors that the potential limitation of this work is the exact dependence of norm $B$ is not justified. It would be interesting future work to find the tightest $\ell_2$ error bound of the SIMs.

---

> ### Author Rebuttal · Authors · 2023-08-09
>
> We wish to thank the anonymous reviewer for their comments.
>
>   Regarding the significance of our Theorem 4.1, we stress that our distributional assumption (subgaussian concentration) is significantly milder than assuming the marginal to be bounded (e.g., a bounded marginal is, in particular, doubly exponentially concentrated). In fact, if we assume that the marginal is bounded and follow the same approach through distortion bounds, we obtain the same ($O(\mathrm{opt})$) guarantee as the one in [FCG20] for the sigmoid (our bounds improve from $\tilde{O}(\mathrm{opt})$ to $O(\mathrm{opt})$).
>
> In our presentation, we emphasized subgaussian distributions to highlight that our method can handle broad classes of unbounded distributions.
>
>   Moreover, Theorems 4.1 and 4.3 provide new guarantees for the standard algorithm of logistic regression (i.e., minimizing the matching loss corresponding to the logit link) through a simple approach (distortion inequalities). Interestingly, we give the best known guarantees for the case where we only assume subgaussian concentration for the marginal (and no anti-concentration or anti-anti-concentration).
>
>   The algorithm we use is the one proposed in [GHK+23], as is implied within our proofs. In particular, the algorithm computes a calibrated and multiaccurate predictor by running a large enough alternating sequence of multiaccuracy and calibration steps (see Algorithm 2 in [GHK+23]). We are going to provide the pseudocode of the algorithm in the final version of the paper as it will indeed be helpful for the reader.
>
> Our results are focused on providing accurate performance guarantees rather than optimizing the sample or time complexities. For this reason as well as for ease of presentation, we decided to not include the exact quantitative bounds on the sample and time complexity of our algorithm (although we do mention that they scale polynomially in all relevant parameters). However, the bounds are implicit in our analysis and we could extract them in case the reviewers believe this is important.
>
> Regarding the question of the reviewer about line 261, if we assume that $g’(s)\not\in [-1,2]$ for any $|s|\le 2\lambda B^2$, we have
> $$\mathrm{opt}_g = \mathbb{E}[(g’(w^*\cdot x) - y)^2] \ge
> \mathbb{E}[(g’(w^*\cdot x) - y)^2 \|\ |w^*\cdot x| \le 2\lambda B^2] \cdot Pr[|w^*\cdot x| \le 2\lambda B^2] \ge 1 \cdot Pr[|w^*\cdot x| \le 2\lambda B^2],$$
> since $y\in [0,1]$.
>
>
>   While multiaccuracy and calibration are indeed relevant to our results, we primarily focus on the use of existing results from the literature of Omniprediction [GHK+23] (where multiaccuracy, multicalibration and calibration notions are presented and studied thoroughly) to acquire bounds for agnostic learning through the idea of distortion inequalities. In particular, [GHK+23] establish a non-trivial connection between fairness and omniprediction, while we focus on establishing a non-trivial connection between omniprediction and classical agnostic learning.
>
> In general, a calibrated predictor $p(x)$ has the property that conditioning $x$ on any of the level sets of $p$ (so that $p(x) = v$), the expected value of $y$ is close to the value of $p(x)$($=v$). A multiaccurate predictor is a predictor that is accurate in expectation with respect to its correlation to any function within a given concept class. A predictor that is calibrated and multiaccurate is an omnipredictor with respect to all matching losses corresponding to Lipschitz links (i.e., it minimizes all these matching losses simultaneously – see also lines 108-110 and Theorem 3.2 where the algorithm essentially computes a calibrated an multiaccurate predictor). Finally, as we mention in lines 103-105, the stationary points of a matching loss (points with zero gradient) correspond to multiaccurate predictors (see also Theorem 5.6 of [GHK+23]).

---

> > ### Comment · Reviewer_uXX7 · 2023-08-15
> >
> > I think the authors addressed my questions properly. I think agnosticly learning SIMs is an interesting problem and I would like to see this paper published after refinement. I have changed my grade from 4 to 6.

---

### Official Review · Reviewer_6FFA · 2023-07-07

**Soundness:** 4 excellent
**Presentation:** 2 fair
**Contribution:** 4 excellent
**Rating:** 8
**Confidence:** 3

**Summary:**

The paper gives an efficient algorithm for learning Single Index Models with arbitrary monotone and Lipschitz function under the condition the marginal distribution of x has bounded variance in all directions.

The error guarantee of the algorithm is $O(B \sqrt{\lambda} \sqrt{\mathrm{opt}})+\epsilon$ ($B=\|w\|_2$ and $\lambda$ is the bound on variance) which is weaker than some previous result but this paper also has less assumption on the marginal distribution.

The authors give an SQ lower bound justifying the dependence of their error on B. However, that lower bound does not imply the error need to have a polynomial dependence on B (i.e. a $\exp (\log^{1/2} B$) dependence should be suffice to not contradict the lower bound).

The high level idea of the algorithm is this:
The authors considers matching loss as the surrogate loss function.
In the agnostic setting, it no longer holds that the matching loss and
Squared loss has the same minimizer.
Instead the authors observes that there is a “distortion bound” between matching loss (for bi-Lipschitz function $u$) and squared loss (lemma 2.2).
Namely, this means for any prediction $p$ and true value $y$, the matching loss is at most off by some offset (which is the matching loss between $y$ and itself) and then a multiplicative factor (which depends on the bi-Lipschitz parameter of $u$).
This implies small matching loss implies small squared loss.
(Up to some $O(\mathrm{opt})$ error.)

Then the argument follows that in order to learn a SIM with an unknown activation function $u$.
We can use existing result to learn a “omnipredictor” that has small matching loss for any bi-Lipschitz function.
Let $u’$ be nearest bi-Lipschitz function of $u$.
Due to the boundedness, it suffices to consider $u’$ instead of $u$.
This gives their algorithm for SIM.

Some other result they show (using similar idea) are (ignoring the dependence on $B$ in the error):
1. An distribution independent efficient algorithm for leaning GLM up to error $O(\mathrm{opt})$ when the activation function is bi-Lipschitz
2. An efficient algorithm for logistic regression up to $O(\mathrm{opt})$ under some concentration assumption of the distribution

**Strengths:**

The algorithm holds under very mild assumption on the marginal distribution.
The distortion bound the authors prove might also have other interesting applications.

**Weaknesses:**

The main weakness is the authors do not have a lower bound that can match the upper bound result they give.
It would be interesting to see if one can get a better algorithm or prove a tighter lower bound.

**Questions:**

It would be better if the authors can have a paragraph summarizing the high level ideas of the algorithms.

**Limitations:**

The authors have adequately addressed the limitations.
There is potential negative societal impact.

---

> ### Author Rebuttal · Authors · 2023-08-09
>
> We wish to thank the anonymous reviewer for their suggestions and for appreciating our results. The reviewer is right that it would be interesting to have tight results (at least in the statistical query framework). However, our work not only provides the first upper bound for learning SIMs in the agnostic setting, but also demonstrates a link between the literature of omniprediction (which is in turn connected to notions from fairness like calibration and multicalibration) and the problem of learning SIMs, which might be useful in proving better upper or lower bounds in future work.

---

> > ### Comment · Reviewer_6FFA · 2023-08-18
> >
> > Thankyou for the clarification.

---

### Official Review · Reviewer_Buwi · 2023-07-07

**Soundness:** 4 excellent
**Presentation:** 3 good
**Contribution:** 3 good
**Rating:** 5
**Confidence:** 4

**Summary:**

This paper studies the learning of Single-Index Models (SIMs) with arbitrary monotone and Lipschitz activations. In SIM model, labeled examples $(x, y)$ are assumed to satisfy $E[y|x] = u^{-1}(w.x)$, where $w$ is an unknown vector, and  $u$ is an unknown monotone function (a.k.a. link function). Given IID-drawn samples from an unknown distribution $D$ over $\mathbb{R}^d\times [0,1]$, the learner's objective is to minimize a predefined loss (e.g., square loss) among the class of SIM models. This paper presents a learning algorithm with a square loss $O(\sqrt{opt})+\epsilon$ if $u^{-1}$ is 1-Lipschitz, where $opt$ is the minimum square error among all SIM models with bounded-norm  $w$.  Moreover, the authors provide related results for standard algorithms like GLMtron and logistic regression.

**Strengths:**

The paper is solid, and the proof techniques are interesting. This paper considers a harder problem of the non-realizable case ($opt \neq 0$) and unknown activation function. Moreover, the analysis is based on more relaxed conditions compared to prior works as it requires the marginals to have bounded second moments.

**Weaknesses:**

The paper claims (in the title and abstract) to prove agostic learnability. This is misleading! First, agnostic in the PAC learning context implies no distributional assumptions on the labeled samples, but a bounded second moment is assumed in this work.
Second, the agnostic learner has an error $opt+\epsilon$ which is clearly not the case here! The bound in Theorem 3.1 is of the form $O(B\sqrt{opt})+\epsilon$, where $B$ is the bound on $||w||$.
Given that the paper shows "weak learnability" as best.

The paper is well-written in general but there are several typos that need to be corrected:
- The notation for sphere, $\mathbb{S}^{d-1}$, is not defined.
- Line 158: is $f'$ the derivative of $f$?
- Line 169: $d$ is missing:  $c:\mathbb{R}\to \mathbb{R}$ need to be  $c:\mathbb{R}^d\to \mathbb{R}$
- Line 180: $d$ is also missing.


I think it would be better if Definition 1.4 and 1.5 were written as Assumptions.


**Questions:**

Q1. When do we get the $opt+\epsilon$ error?

Q2. What about other loss functions beyond the square error and logistic loss?

Q3. The bound for logistic loss scales with B as $e^{B^2}$, isn't that problematic?


**Limitations:**

See weaknesses.

---

> ### Author Rebuttal · Authors · 2023-08-09
>
> We thank the anonymous reviewer for their comments and for appreciating our work.
>
> You make the point in your review that ‘agnostic learning’ should mean ‘distribution-free agnostic learning,’ as the term was defined in the 1994 Kearns Schapire paper that originally defined the model.   We would like to stress that the way we (and others) use the term ‘agnostic learning’ to mean ‘distribution-specific agnostic learning’ is now standard and has widely appeared as such in the literature over the past two decades. More concretely, the two concerns of the reviewer about our use of “agnostic learning” were the following 1) we make an assumption on the marginal distribution of the features (note that we do not make such an assumption on the labels) and 2) the form of the error guarantee. Both of these restrictions are now common in the (classical and recent) agnostic learning literature. For example, see Theorem 3 in the classical paper [1] on agnostic learning of halfspaces (note that the algorithm is still called an agnostic learner) and the more recent paper [2] where they propose an agnostic learning algorithm for a single ReLU neuron with guarantee $\mathrm{poly}(\mathrm{opt})$, under some assumptions about the marginal distribution. Note that both of these papers also use the term agnostic learning in their title. The term is generally used to emphasize that the learner is agnostic to how the *labels* are generated and is not tied to a specific learning scenario.
>
>   We also thank the reviewer for pointing out some typos, which we will fix in future versions of the paper.
>
> Regarding the questions posed:
>
> Q1. There have been (statistical query) lower bounds in prior work (see lines 127-130) that provide evidence for the hardness of learning up to $\mathrm{opt} + \epsilon$, even when the activation is fixed to be a ReLU and the marginal is the standard Gaussian. Although we have already provided pointers to such lower bounds, we will exploit the additional space provided for the camera-ready version and add more details in Section 1.1.
>
> Q2. We focus on the squared error, which is a standard loss function used in learning theory and provides results comparable to prior work in the field (e.g., [2]). However, our technique is general; our analysis is based on the idea of translating bounds with respect to matching losses to results with respect to the squared loss and it is conceivable that we could obtain results even when we substitute the squared loss with some other loss function of interest, by proving corresponding distortion bounds. We note that this might entail some technical complications, as the squared loss is usually more analytically convenient than other losses.
>
> Q3. The bound we provide for the squared error of the logistic regression algorithm scales indeed exponentially with the bound on the norm of the parameter vector. However, please keep in mind:
>
> 1. Some dependence on the norm is expected (since we do not make any anti-concentration assumptions on the marginal) and the result is useful when the norm bound is constant.
>
> 2. Prior work contains bounds that are quantitatively similar (with even more assumptions taken). In Theorem 3.3 of FCG [2], where it is assumed that the marginal distribution is in fact bounded, the dependence on the norm bound is also exponential (since their parameter $\gamma$ in their Assumption 3.1 would decay exponentially with $\rho$ when $\sigma$ is the sigmoid). Note that our method can also yield an $O(\mathrm{opt})$ result for learning the sigmoid if we assume the marginal to be bounded.
>
> 3. Our result demonstrates that a standard algorithm (minimizing logistic loss) achieves guarantees that are state-of-the-art at least in some regime.
> 4. We once more use our simple analysis approach based on distortion bounds.
>
> [1] Kalai, A.T., Klivans, A.R., Mansour, Y., & Servedio, R.A. (2005). Agnostically learning halfspaces. FOCS 2005, SICOMP 2008.
>
> [2] Frei, Spencer, Yuan Cao, and Quanquan Gu. Agnostic learning of a single neuron with gradient descent. NeurIPS 2022.

---

### Official Review · Reviewer_VkT9 · 2023-07-09

**Soundness:** 3 good
**Presentation:** 1 poor
**Contribution:** 3 good
**Rating:** 4
**Confidence:** 1

**Summary:**

This work studies the problem of agnostically learning single index models with arbitrary Monotone and Lipschitz activations. Compared prior work, this work establishes the existence of an learning algorithm under more relaxed assumptions. This work is based on recent work by Gopalan et al. [2023] on Omniprediction using predictors satisfying calibrated multiaccuracy.

**Strengths:**

This paper studies a difficult but realistic setting, and the topic of agonistically learning single index model is very interesting and important in itself.

**Weaknesses:**

I find this paper a bit difficult to read for general researchers in the machine learning community, especially those who are not familiar with the related literature.

I also have a (perhaps naive) question about the claims in the abstract and line 33. It seems to claim that this paper devises an "algorithm" that can efficiently learn SIM in an agnostic setting, I wonder what is the algorithm actually? It seems all theoretical results in this paper are all pointing to the existence of such an algorithm, but not an actual algorithm that can be used to estimate it.

**Questions:**

My questions are raised in the previous section.

**Limitations:**

This paper may benefit from polishing up the write-up and make it more friendly for general ML researchers or practitioners.

---

> ### Author Rebuttal · Authors · 2023-08-09
>
> We thank the anonymous reviewer for expressing their concerns about the readability of our paper.
>
> As we stated in the global response, we are planning to make certain modifications which we believe will significantly improve the readability of our paper by researchers of diverse backgrounds. The reason we believe so is that there were no objections on the overall structure of our paper and in particular (after resolving a small number of issues pointed to by the reviewers): The introduction section contains minimal technical formalism and should be accessible by researchers of diverse backgrounds. It contains the definition of our setting, its significance, informal versions of our main results (Theorems 1.1, 1.2, 1.3) as well as several pointers to prior work for further reading (e.g., Section 1.1). The following sections of our main paper contain some technical details, but we view this as a strength rather than a weakness, since it contributes to the precision of our statements, which is important for the problem we consider. That said, Section 1.2 provides the required background for the following sections. Hence, Sections 2, 3, 4 and 5 should also be accessible by a diverse audience, although an in-depth approach might be easier for researchers in the respective area.
>
>   Regarding the second question of the reviewer, it is implicit in our proofs that we use the algorithm of Gopalan et al. [2023] for calibrated multiaccuracy (see Section 7.2 of Gopalan et al. [2023]). In particular, the algorithm computes a calibrated and multiaccurate predictor by running a large enough alternating sequence of multiaccuracy and calibration steps (see Algorithm 2 in Gopalan et al. [2023]). The reviewer correctly points out that we should state the algorithm we use explicitly (in pseudocode), which we are planning to do for the camera-ready version of the paper exploiting the additional page allowed for the final version.

---

### Author Rebuttal · Authors · 2023-08-09

We wish to thank the anonymous reviewers for their constructive feedback! In this global response, we provide general responses to concerns shared by more than one of the reviewers and we provide more specific answers in the personal responses.

It is true that our upper and lower bounds on the approximation guarantee are not tight, but we are the first to obtain nontrivial bounds in this very general setting– we succeed simultaneously across all monotone Lipschitz activations.  We also make the first connection between SIM learning and multiaccuracy/omniprediction.  We leave tightening these bounds as an important open problem.

Some of the reviewers expressed some concerns regarding the readability of our paper. Although this was not a concern shared by all of the reviewers, in order to make our work even more accessible to a more diverse audience, we have a concrete plan that consists of valuable but minimal additions and modifications for the final version (exploiting the additional space allowed). In particular, 1) we will provide a larger amount of detail in Section 1.1, so that the scope of our work with respect to the relevant literature is clearer, 2) we will add a description for the algorithm we use (which is based on the algorithm of Gopalan et al. [2023] for calibrated multiaccuracy) in pseudocode and 3) we will correct typos and make clarifications in our proofs according to the reviewers’ suggestions (or any additional suggestions they provide during the discussion period). Overall, we believe that these changes are minimal, but may resolve much of the concerns raised by some of the reviewers, since there were no major objections on the overall structure of our paper or the architecture of our proofs (i.e., hierarchy of claims/lemmas).

One reviewer compared some of our results to the FCG paper “Agnostic Learning of a Single Neuron with Gradient Descent” – please note that the FCG paper has a hidden factor of d inside their O(opt) bound even when the marginal distribution is a spherical Gaussian, yielding a very weak guarantee (additionally they must know the activation $\sigma$ beforehand).  Our bounds only depend on the second moment of the marginal distribution.  If we make the further assumption that the distribution is bounded as FCG does, then we actually obtain the identical O(opt) guarantee as FCG (we give more comments on FCG below).

---

### Decision · Program_Chairs · 2023-09-21

**Decision:**

Accept (poster)

**Comment:**

This paper studies the problem of learning single index models (SIMs) in the agnostic setting. The main result is an efficient algorithm that approximately learns the desired function with respect to a general class of marginal distributions within multiplicative approximation ratio of $O(B^{1/2})$, where $B$ is a bound on the $\ell_2$-norm of the parameter vector. The main technique leverages recent prior work by Gopalan et al. on omnipredictors. The problem studied is fundamental and the underlying results and techniques are interesting. Overall, the paper is a solid contribution to this literature and is appropriate for publication at NeurIPS. Some reviewers had legitimate concerns about the current presentation of the work. The authors are encouraged to revise their work accordingly.